# Infant gut microbiome composition is associated with non-social fear behavior in a pilot study

Alexander L. Carlson [1], Kai Xia[2], M. Andrea Azcarate-Peril[3,4], Samuel P. Rosin [5], Jason P. Fine[5], Wancen Mu[5], Jared B. Zopp[2], Mary C. Kimmel [2], Martin A. Styner [2,6], Amanda L. Thompson[7,8], Cathi B. Propper[1] & Rebecca C. Knickmeyer [2,9,10,11✉]

Experimental manipulation of gut microbes in animal models alters fear behavior and relevant neurocircuitry. In humans, the first year of life is a key period for brain development, the emergence of fearfulness, and the establishment of the gut microbiome. Variation in the infant gut microbiome has previously been linked to cognitive development, but its relationship with fear behavior and neurocircuitry is unknown. In this pilot study of 34 infants, we find that 1-year gut microbiome composition (Weighted Unifrac; lower abundance of *Bacteroides*, increased abundance of *Veillonella*, *Dialister*, and Clostridiales) is significantly associated with increased fear behavior during a non-social fear paradigm. Infants with increased richness and reduced evenness of the 1-month microbiome also display increased non-social fear. This study indicates associations of the human infant gut microbiome with fear behavior and possible relationships with fear-related brain structures on the basis of a small cohort. As such, it represents an important step in understanding the role of the gut microbiome in the development of human fear behaviors, but requires further validation with a larger number of participants.

[1] Frank Porter Graham Child Development Institute, University of North Carolina, Chapel Hill, NC, USA. [2] Department of Psychiatry, University of North Carolina, Chapel Hill, NC, USA. [3] Department of Medicine, University of North Carolina, Chapel Hill, NC, USA. [4] Microbiome Core Facility, University of North Carolina, Chapel Hill, NC, USA. [5] Department of Biostatistics, University of North Carolina, Chapel Hill, NC, USA. [6] Department of Computer Science, University of North Carolina, Chapel Hill, NC, USA. [7] Department of Anthropology, University of North Carolina, Chapel Hill, NC, USA. [8] Department of Nutrition, University of North Carolina, Chapel Hill, NC, USA. [9] Department of Pediatrics and Human Development, Michigan State University, East Lansing, MI, USA. [10] Institute for Quantitative Health Science and Engineering, Michigan State University, East Lansing, MI, USA. [11] C-RAIND Fellow and Co-Director, Michigan State University, East Lansing, MI, USA. ✉email: knickmey@msu.edu

Fear—a behavioral response to threat—is an evolutionarily conserved mechanism that promotes survival. The emergence of fear is an important component of normal human development. Fear behavior in response to different environmental stimuli emerge on a schedule that appears to parallel developmentally-relevant fitness threats, supporting an evolutionary non-associative model of fear acquisition[1,2]. Around 6 months of age, infants demonstrate fear processing through discrimination of fearful faces from other facial expressions[3–5]. Fear of heights, strangers, and strange objects can be reliably detected in the subsequent months[6] and behavioral responses increase in intensity until at least 11 months of age, with many fear-evoking paradigms eliciting peak intensities around 12 months of age[7]. Fear may act as a regulatory or protective mechanism to balance the increase in mobility and exploratory behavior during this time[8]. This general developmental pattern is observed in different individuals, but the intensity of fear in response to a specific threat can vary with high levels of developmental fear predicting the future emergence of anxiety disorders[9,10] while a lack of early fear behavior may be associated with future callous-unemotional traits[11].

The microbiome has been associated with animal behavior across the Chordata phylum[12] and may be one mechanism impacting fear behavior. Several studies have found that administering probiotic species of *Lactobacillus, Bifidobacterium*, or *Bacteroides* yields anxiolytic effects in rodents[13–16]. In germ-free mouse models and a study in germ-free zebrafish, manipulating the gut microbiome has marked effects on fear behavior[17–23]. For example, germ-free mice demonstrate reduced expression of fear behavior compared to conventionally colonized mice in the open field, elevated plus maze, and light-dark box, whereas germ-free rats display increased fear behavior[24]. In non-mammalian vertebrates, germ-free zebrafish display similar reductions in fear behavior compared to conventionalized controls in an open-field analog. The timing of exposure to the microbiome appears to be particularly important in the development of fear behavior. Germ free mice colonized with specific pathogen free microbiota at 3 weeks of age exhibit normal fear behavior. However, when colonized later in development at 10 weeks of age, fear behavior remains abnormal[19,25]. Germ-free animal models are a unique experimental system that do not completely reflect biological reality. Never-the-less, these studies do suggest there are critical neurodevelopmental windows in which individual differences in composition of the microbiome may impact fear behavior acquisition and expression.

Microbiome manipulation in rodent models also results in structural and biochemical changes to brain structures canonically involved in fear circuitry including the amygdala, hippocampus, and medial prefrontal cortex[26–30]. For example, germ-free conditions resulted in changes to the morphology of cells in the amygdala where dendritic length and spine density was increased along with increased overall amygdala volume[31]. There are also broad changes in amygdala microRNA expression along with mRNA expression of NGFI-A, BDNF, and NR2B in germ free conditions[18,20,32]. Similar effects are observed in the hippocampus where germ free mice have larger hippocampal volumes, decreased dendritic length and spines, changes to the hippocampal serotonergic system, and altered mRNA expression[25,31,33,34]. *Lactobacillus* administration reduced $GABA_{A\alpha2}$ receptor mRNA expression in the amygdala and prefrontal cortex and increased expression in the hippocampus[14]. Finally, microbiome colonization is important for normal myelination of the prefrontal cortex[35,36].

The same fear structures impacted by microbiome manipulation in rodents are developmentally important and experience rapid growth and activity during the first year of life in humans. Amygdala, hippocampus, and prefrontal cortex gray matter volumes increase ~105%, 84%, and 104% from birth to 1 year of age, respectively[37]. Cellular processes underlying gross volumetric changes include myelination of axons, synaptogenesis, proliferation and migration of glia, and differentiation of oligodendrocytes[38–40]. The connectivity of these structures—especially bottom-up signaling from amygdalae to prefrontal cortex—are crucially important for early affective learning and behavior during infancy and childhood[41]. In addition, early-life stressors have been shown to have long-term consequences on amygdala function and anxious behavior[42]. Given the structural growth and long-term consequences of perturbations occurring during this time, the first year of life may represent a critical period for development of fear circuitry.

The studies reviewed above suggest that the infant gut microbiome may be a key component in the normal development of fear behavior and associated brain structures. In addition, it may have a role in subsequent psychopathology, since fear behavior represents a fundamental behavioral dimension that is disrupted in multiple psychiatric disorders. Altered microbial composition has been reported in individuals with generalized anxiety disorder, autism, depression, schizophrenia, attention deficit hyperactivity disorder (ADHD), and eating disorders[43–47]. A meta-analysis of probiotic interventions[48] suggests positive effects on anxiety symptoms, and transplant of microbial communities from people with psychiatric illnesses into rodent models can induce relevant behaviors not seen when microbial communities from healthy controls are transplanted[22,49–53]. However, to date, there have been no studies testing the association of the gut microbiome with observed fear behavior and fear-related brain structures during early human development.

For this pilot study, we recruited a prospective longitudinal cohort of 34 infants and followed them from 1 month to 1 year of age. These timepoints were selected because they index key periods in the development of the microbiome, brain, and fear behavior. The developmental phase of the microbiome spans these ages[54], brain volumes show the greatest developmental change[55], and peak reactivity to fear-inducing stimuli occurs around 1 year of age[7]. To reduce confounding effects of birth mode, antibiotic exposure, and feeding practices, all infants were vaginally delivered, antibiotic naïve, and exclusively breastfed until their first study visit at 1 month. In this study, we tested whether features of 1-month and 1-year gut microbiomes were associated with the volume of brain regions involved in fear circuitry at each age as well as fear behaviors at 1 year of age. Specifically, we examined behavioral inhibition, the tendency to display fear and withdrawal when presented with novel stimuli or situations. Behavioral inhibition measured in infancy has been shown to predict internalizing psychopathology in adulthood[10]. Traditionally, behavioral inhibition has been treated as a unitary construct, but recent work suggests it may be useful to distinguish between reactions to non-social and social stimuli which have differing implications for predicting later pathology[56]. We hypothesized that infants with relatively greater amounts of microbiota linked to reduced fear in mice, specifically *Lactobacillus*[13], *Bifidobacterium*[15] and *Bacteroides*[16,57], would exhibit reductions in both non-social and social fear. Similarly, we predicted the same infants would show volumetric differences in three structures involved in fear circuitry: the amygdala, hippocampus, and medial prefrontal cortex.

In this work we show that that 1-year gut microbiome composition (Weighted Unifrac; lower abundance of *Bacteroides*, increased abundance of *Veillonella*, *Dialister*, and Clostridiales) is significantly associated with increased fear behavior during a non-social fear paradigm. Infants with increased richness and reduced evenness of the 1-month microbiome also display increased non-social fear. No relationship is observed with social

| | Facial fear | Vocal distress | Bodily fear | Startle | Escape behavior | Wariness | IBQ-R fear |
|---|---|---|---|---|---|---|---|
| **Table 1 Spearman correlations between fear behavior outcomes at 1 year of age.** | | | | | | | |
| Facial fear | — | 0.98 | 0.87 | 0.80 | 0.52 | −0.04 | 0.05 |
| Vocal distress | — | — | 0.88 | 0.80 | 0.59 | −0.01 | 0.13 |
| Bodily fear | — | — | — | 0.71 | 0.56 | 0.10 | −0.08 |
| Startle | — | — | — | — | 0.32 | −0.03 | 0.01 |
| Escape behavior | — | — | — | — | — | −0.08 | 0.26 |
| Wariness | — | — | — | — | — | — | 0.04 |
| IBQ-R fear | — | — | — | — | — | — | — |

Facial fear, vocal distress, bodily fear, startle, and escape behavior are measured during the Mask Task. Wariness is measured during the Strange Situation Paradigm. The IBQ-R scale is a parent report questionnaire measure, the fear subscale was used. Bold formatting denotes correlations with $p < 0.05$. Vocal distress and facial fear ($p = 1.104e-13$), bodily fear and facial fear ($p = 6.763e-07$), bodily fear and vocal distress ($p = 3.356e-07$), startle and facial fear ($p = 2.243e-05$), startle and vocal distress ($p = 2.785e-05$), startle and bodily fear ($p = 0.0004$), escape behavior and facial fear ($p = 0.017$), escape behavior and vocal distress ($p = 0.006$), escape behavior and bodily fear ($p = 0.01$). Source data for this table are provided as a Source Data file.

fear behavior or parental report of fear behavior. Finally, non-significant trends are observed between the microbiome and volumes of the amygdala and medial prefrontal cortex, brain regions with important roles in processing threat.

## Results

**1-month and 1-year microbiome associated with non-social fear behavior.** Two observational behavioral assessments, the Mask Task and Strange Situation, were chosen to assess non-social and social fear, respectively. In addition, parents completed the revised Infant Behavior Questionnaire (IBQ-R) which includes a range of fear behaviors both social and non-social. Measures of facial fear, bodily fear, vocal distress, escape behavior, and startle within the Mask Task paradigm were highly correlated, suggesting they tap into the same underling behavioral construct. However, there was little correlation between Mask Task measures, social wariness during the Strange Situation paradigm, and parent report measures of fear on the IBQ-R, suggesting these measures capture different behavioral constructs (Table 1). Behavioral outcomes were tested for associations with 1-month ($n = 32$) and 1-year ($n = 21$) microbiome community measures of alpha and beta diversity. Alpha diversity, the diversity of the microbiome within the individual, was analyzed through a principal component analysis of four alpha metrics (Shannon Diversity, Observed Species, Faith's Phylogenetic Diversity, Chao1) to generate Alpha Diversity PC1 and Alpha Diversity PC2 at 1 month and 1 year of age (Fig. 1). At 1 month, Alpha Diversity PC1 appears to captures richness, while Alpha diversity PC2 captures evenness. At 1 year, Alpha Diversity PC1 appears to capture species richness, while Alpha Diversity PC2 captures both evenness and taxonomic richness; Beta diversity, the dissimilarity of the microbiome between individuals, was generated separately at 1 month and 1 year of age through principle coordinate analysis of Weighted Unifrac to generate principal coordinates 1 and 2 at each age.

To better understand how community composition influences beta diversity principal coordinates, we calculated correlation coefficients between the relative abundance of each genus and each principal coordinate. At 1 month, Weighted Unifrac PC 1 had strong positive associations with *Bifidobacterium* and *Streptococcus* and strong negative associations with *Bacteroides*. PC2 at 1 month of age had strong positive associations with *Veillonella* and negative associations with an unnamed genus of Enterobacteriaceae and *Bifidobacterium* (Fig. 2). At 1 year, Weighted Unifrac PC 1 had strong positive correlations with *Bacteroides* and strong negative correlations with *Veillonella*, *Dialister*, an unnamed genus of Clostridiales, *Bifidobacterium*, and *Lactobacillus*. PC2 at 1 year of age had strong positive

associations with *Bacteroides* and *Dialister* and strong negative associations with *Bifidobacterium* (Fig. 3).

Linear mixed effect models with random intercept were used to test for effects of alpha and beta diversity on repeated measures outcomes of non-social fear behavior and social fear behavior. Multiple linear regression models were used for the IBQ-R fear index. For the Mask Task non-social fear paradigm, up to four different masks were presented to the infant with facial fear, vocal distress, bodily fear, startle response, and escape behavior as fear outcomes for each mask. Each of these outcomes were highly correlated and also repeated across the four possible mask presentations. To accommodate this, we used a two-level mixed effects model with Bonferroni correction to account for multiple testing. Bonferroni significance for behavioral testing was set at $p = 0.00208$ for 24 tests (4 microbiome measures (Weighted Unifrac PC 1 & 2, Alpha PC 1 & 2) and the number of models that were run (2 models for the Mask Task, 2 for the Strange Situation paradigm, and 2 for the IBQ-R, with one model examining associations with the 1-month microbiome and one model examining associations with the 1-year microbiome).

There was a significant negative association between 1-month alpha diversity principal component 2 and non-social fear behaviors across the total number of masks presented ($p = 0.00038$, $n = 19$) (average score used for visual representation in Fig. 4). Other 1-month microbiome measures of alpha and beta diversity showed no significant associations with Mask Task outcomes after Bonferroni correction. 1-year beta diversity measure, Weighted Unifrac PC 1, had a significant negative association with non-social fear behaviors across the total number of masks presented in the Mask Task paradigm ($p = 0.00010$, $n = 14$) (average score used for visual representation in Fig. 5). Other 1-year microbiome measures of alpha and beta diversity were not significantly associated with Mask Task outcomes after Bonferroni correction.

The alpha and beta diversity measures used in this study capture complex patterns of variation within microbial communities; to better understand how specific bacterial groups might relate to fear reactivity, we performed a secondary analysis to determine associations with individual genera. Although no relationships were significant after FDR correction, we found that increasing non-social fear behavior in the Mask Task paradigm was associated ($p < 0.05$) with increasing *Dialister* (escape behavior), an unnamed genus in the Clostridiales order (vocal distress, startle), members of the Clostridiales order that could not be confidently assigned to a family or genus (escape behavior), unnamed genus of Erysipelotrichaceae (escape behavior), and *Sutterella* (startle) (see Supplementary Table 2 for further reference). Three of these taxa (*Dialister*, unassigned Clostridiales order, and the unnamed genus of

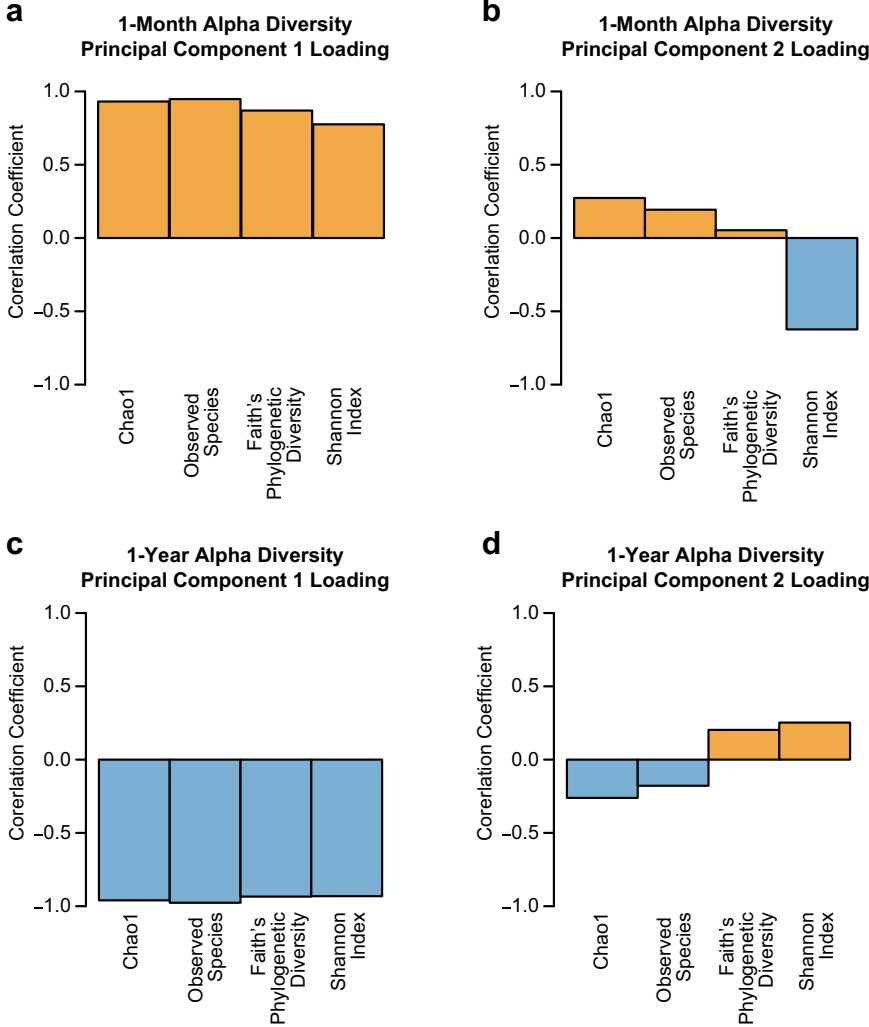

**Fig. 1 Loading of individual alpha diversity metrics on principal components. a, b** Loading of Chao1, Observed Species, Faith's Phylogenetic Diversity, and Shannon Index on alpha diversity principal component 1 and 2 at 1 month of age. **c, d** Loading of Chao1, Observed Species, Faith's Phylogenetic Diversity, Shannon Index on alpha diversity principal component 1 and 2 at 1 year of age. Source data for this figure are provided as a Source Data file.

Erysipelotrichaceae) also showed strong negative associations with 1-year Weighted Unifrac PC1 (Fig. 3).

Social fear assessed through Episode 3 of the Strange Situation paradigm as well as parental report of fear behavior with IBQ-R fear index were not significantly associated with any microbiome measures at 1 month or 1 year of age after Bonferroni correction.

**Infant microbiome and fear-related brain structures**. Neuroimaging was conducted at both study visits to assess regions involved in fear circuitry. Using multiple linear regression models with age at scan and sex as covariates, we tested for associations of alpha and beta diversity with a priori brain volume regions of interest involved in fear circuitry: specifically, medial prefrontal cortex, amygdala, and hippocampus. While none reached the level of significance after Bonferroni correction, 1-month Weighted Unifrac PC 1 was negatively associated with 1-year medial prefrontal cortex volume ($p = 0.046$, $n = 14$) and 1-year Weighted Unifrac PC 1 was negatively associated with 1-year amygdala volume ($p = 0.034$, $n = 13$) (Fig. 6).

As with our investigation of microbiome effects on fear behavior, we performed a secondary analysis to determine associations with individual genera. There was a significant negative association between *Streptococcus* relative abundance at 1 month with amygdala volume at 1 month of age after FDR correction ($q = 0.021$) (Supplementary Table 3). *Streptococcus* was also negatively associated with 1-month hippocampus and medial prefrontal cortex volumes but does not reach FDR significance ($p = 0.009$, $p = 0.012$, respectively). Several other genera at 1 month of age were associated with amygdala and medial prefrontal cortex volumes ($p < 0.05$) at 1 month (Supplementary Table 3) and 1 year of age (Supplementary Table 4). There were no significant associations with 1-year genera and brain volumes at 1 year of age.

**Principal coordinates and their functional capacity**. To better understand how beta diversity community measures might influence brain development, we also tested each principal coordinate for associations with predicted metagenomic functional ability as determined by Phylogenetic Investigation of Communities by Reconstruction of Unobserved States (PICRUSt)[58]. The PICRUSt pipeline predicts the abundance of Kyoto Encyclopedia of Genes and Genomes (KEGG) orthologs by participant at the collapsed KEGG pathway levels. 1-month Weighted Unifrac PC1 was significantly associated with several KEGG pathways after FDR correction ($n = 32$) (see Table 2). There were no significant associations with 1-month Weighted Unifrac PC2, 1-year Weighted Unifrac PC 1, or 1-year Weighted Unifrac PC 2 after FDR correction.

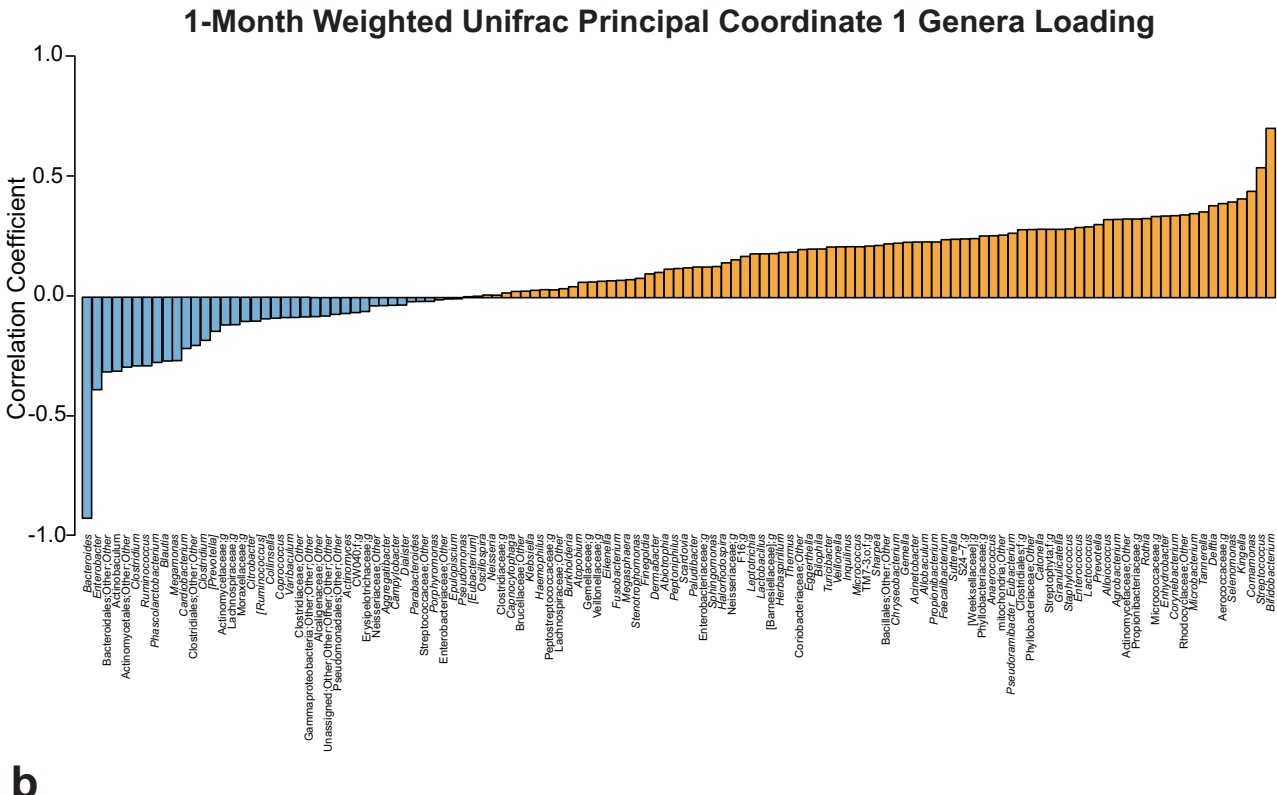

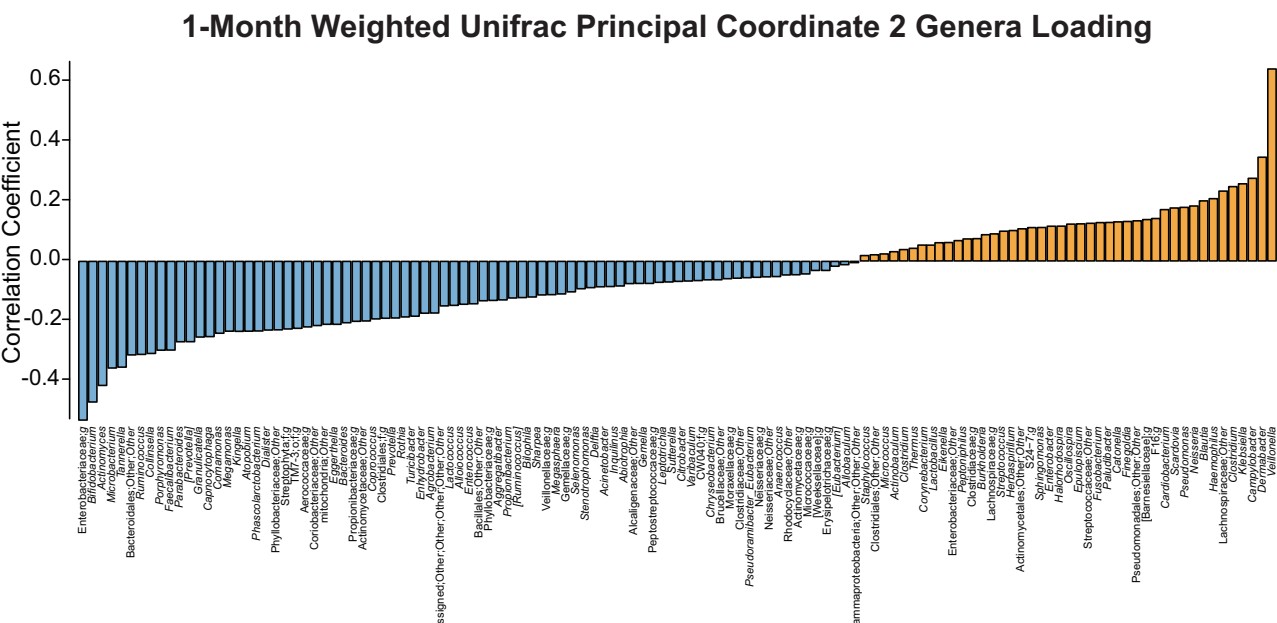

**Fig. 2 Genera correlations with weighted unifrac principal coordinates at 1 month. a** Positive (orange) and negative (blue) correlations between genera and beta diversity metric, Weighted Unifrac principal coordinate 1 at 1 month of age. **b** Positive (orange) and negative (blue) correlations between genera and beta diversity metric, Weighted Unifrac principal coordinate 2 at 1 month of age. Source data for this figure are provided as a Source Data file.

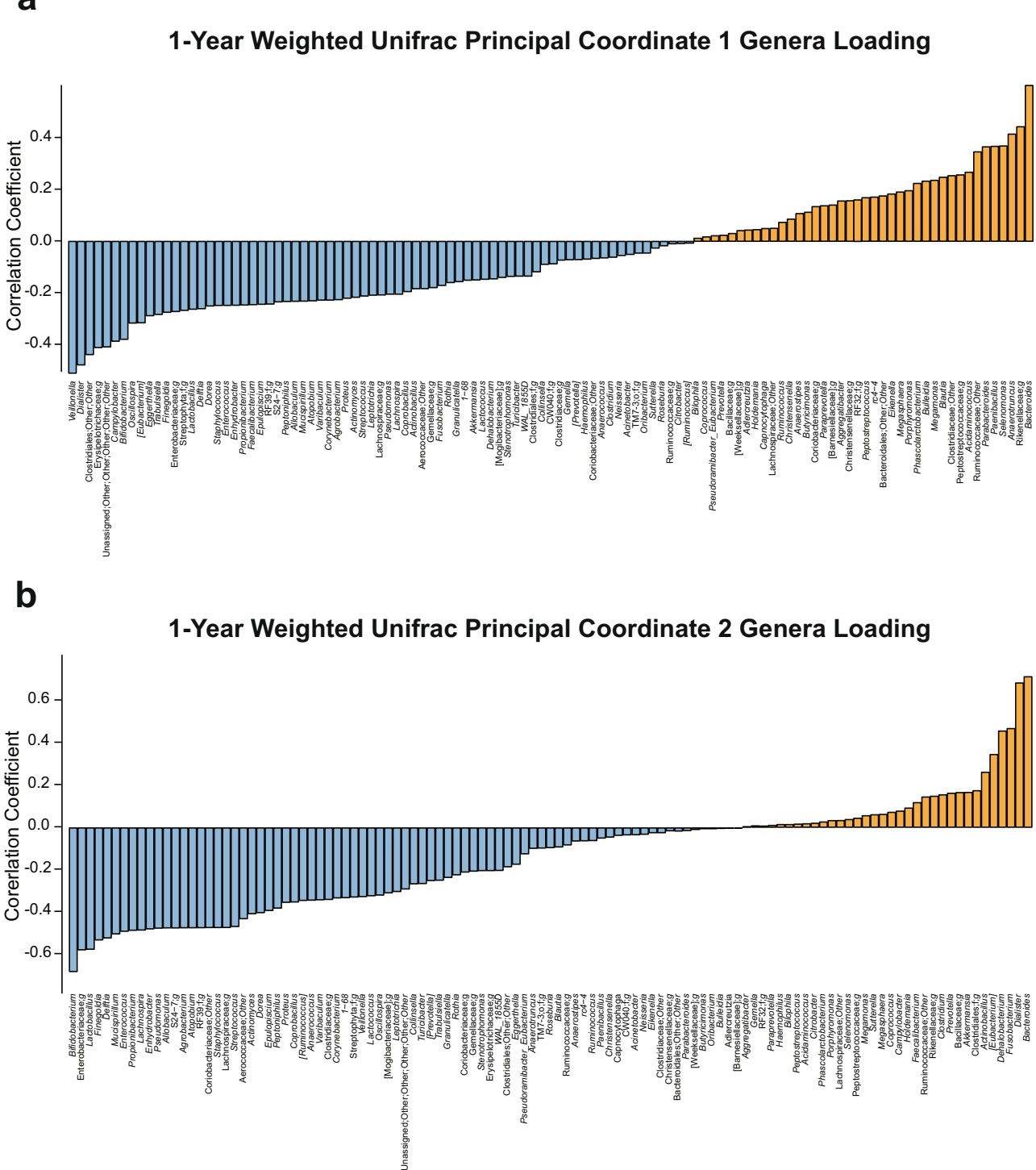

**Fig. 3 Genera correlations with weighted unifrac principal coordinates at 1 year. a** Positive (orange) and negative (blue) correlations between genera and beta diversity metric, Weighted Unifrac principal coordinate 1 at 1 year of age. **b** Positive (orange) and negative (blue) correlations between genera and beta diversity metric, Weighted Unifrac principal coordinate 2 at 1 year of age. Source data for this figure are provided as a Source Data file.

**Longitudinal changes in microbiome from 1 month to 1 year.** Alpha diversity measures, Chao1, Observed Species, Shannon Diversity, and Faith's Phylogenetic Diversity, increased from 1-month to 1-year. Participants with high alpha diversity at 1-month showed the least change while those with low alpha diversity initially had a large increase to 1 year (Supplementary Fig. 1). Similarly, a significant negative relationship was observed between 1-month Shannon Diversity and 1-year Chao1, Observed

Species, and Faith's Phylogenetic Diversity as well as 1-month Observed Species and 1-year Chao1 (Supplementary Table 1). Weighted Unifrac principal coordinates were tested for correlation within and between timepoints and none reached the level of significance.

**Identification of microbiome covariates.** The significant effects we observed could indicate a causal role for the microbiome in

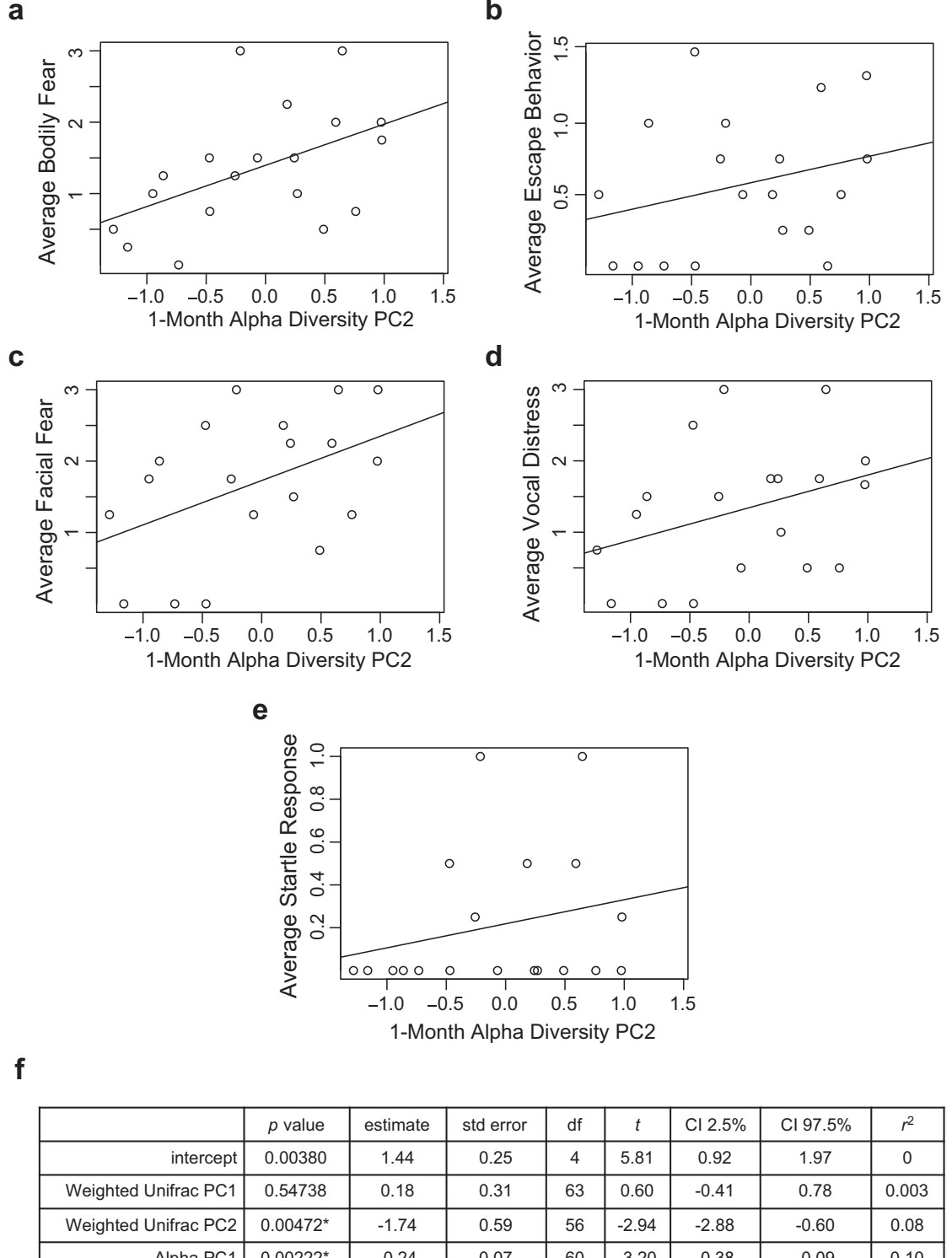

| | $p$ value | estimate | std error | df | $t$ | CI 2.5% | CI 97.5% | $r^2$ |
|---|---|---|---|---|---|---|---|---|
| intercept | 0.00380 | 1.44 | 0.25 | 4 | 5.81 | 0.92 | 1.97 | 0 |
| Weighted Unifrac PC1 | 0.54738 | 0.18 | 0.31 | 63 | 0.60 | -0.41 | 0.78 | 0.003 |
| Weighted Unifrac PC2 | 0.00472* | -1.74 | 0.59 | 56 | -2.94 | -2.88 | -0.60 | 0.08 |
| Alpha PC1 | 0.00222* | -0.24 | 0.07 | 60 | -3.20 | -0.38 | -0.09 | 0.10 |
| Alpha PC2 | 0.00038** | 0.47 | 0.13 | 60 | 3.77 | 0.23 | 0.72 | 0.12 |
| episode | 0 | 0.34 | 0.04 | 198 | 8.22 | 0.25 | 0.41 | 0.15 |

**Fig. 4 Associations between 1-month microbiome and non-social fear behavior. a–e** Scatterplots displaying relationships between alpha diversity principal component 2 at 1 month of age and average scores of bodily fear, escape behavior, facial fear, vocal distress, and startle response during the Mask Task paradigm at 1 year of age, respectively. Each dot represents a single subject. **f** Two-level linear mixed effects model with $t$ test, **denotes significance after Bonferroni correction for multiple comparisons, threshold for significance of $p = 0.00208$, * denotes $p < 0.05$, $n = 19$ independent participants. Source data for this figure are provided as a Source Data file.

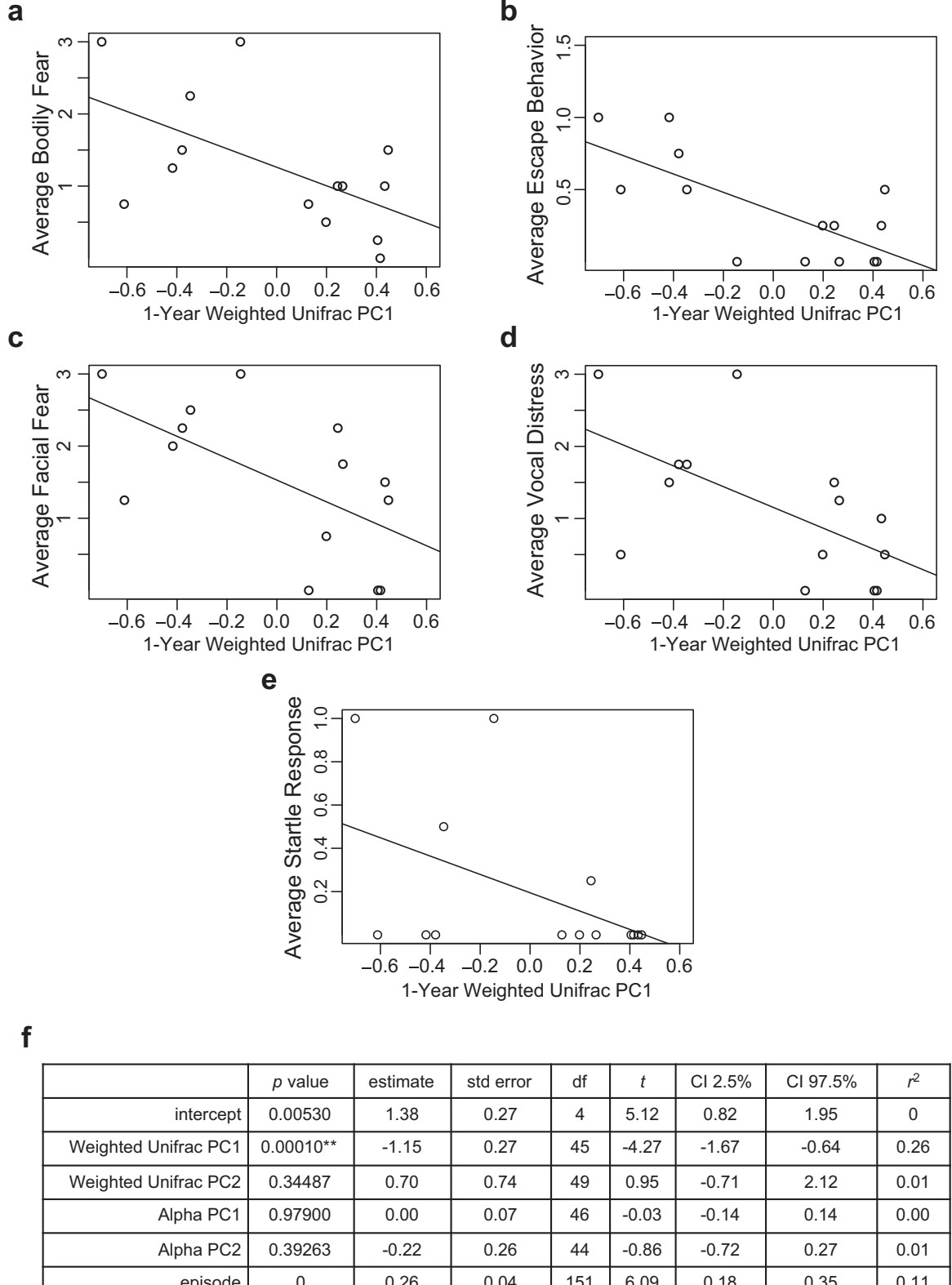

**Fig. 5 Associations between 1-year microbiome and non-social fear behavior. a**–**e** Scatterplots displaying relationships between Weighted Unifrac principal coordinate 1 at 1 year of age and average scores of bodily fear, escape behavior, facial fear, vocal distress, and startle response during the Mask Task paradigm at 1 year of age, respectively. Each dot represents a single subject. **f** Two-level linear mixed effects model with *t* test, **denotes significance after Bonferroni correction for multiple comparisons, threshold for significance of *p* = 0.00208, *denotes *p* < 0.05, *n* = 14 independent participants. Source data for this figure are provided as a Source Data file.

**a**

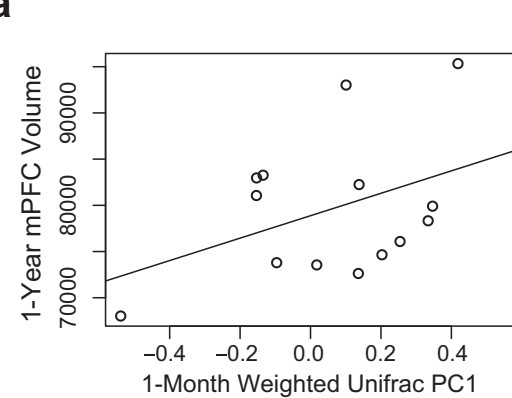

**b**

|  | p value | estimate | std error | df | t | CI 2.5% | CI 97.5% | r² |
|---|---|---|---|---|---|---|---|---|
| intercept | 0.00005 | 65067 | 7314 | 7 | 8.90 | 47772.3 | 82362 | 0.00 |
| Weighted Unifrac PC1 | 0.046* | 22701 | 9389 | 7 | 2.42 | 499.2 | 44904 | 0.34 |
| Weighted Unifrac PC2 | 0.359 | 19073 | 19450 | 7 | 0.98 | -26918.5 | 65064 | 0.06 |
| Alpha PC1 | 0.893 | -168 | 1210 | 7 | -0.14 | -3028.5 | 2692 | 0.00 |
| Alpha PC2 | 0.477 | -2244 | 2989 | 7 | -0.75 | -9310.6 | 4823 | 0.02 |
| Age at 1 yr MRI | 0.059 | 517 | 229 | 7 | 2.26 | -24.7 | 1059 | 0.19 |
| Sex | 0.385 | -3601 | 3885 | 7 | -0.93 | -12788.8 | 5587 | 0.02 |

**c**

**d**

|  | p value | estimate | std error | df | t | CI 2.5% | CI 97.5% | r² |
|---|---|---|---|---|---|---|---|---|
| intercept | 0.00077 | 2012.6 | 321.5 | 6 | 6.26 | 1225.9 | 2799.4 | 0.00 |
| Weighted Unifrac PC1 | 0.0336* | -201.3 | 73.4 | 6 | -2.74 | -380.8 | -21.7 | 0.29 |
| Weighted Unifrac PC2 | 0.785 | 62.0 | 217.9 | 6 | 0.28 | -471.1 | 595.1 | 0.00 |
| Alpha PC1 | 0.223 | -22.8 | 16.8 | 6 | -1.36 | -63.9 | 18.3 | 0.09 |
| Alpha PC2 | 0.652 | -31.3 | 66.1 | 6 | -0.47 | -193.0 | 130.4 | 0.01 |
| Age at 1 yr MRI | 0.925 | 0.08 | 0.8 | 6 | 0.10 | -2.0 | 2.1 | 0.00 |
| Sex | 0.076 | -133.3 | 62.3 | 6 | -2.14 | -285.6 | 19.1 | 0.18 |

**Fig. 6 Associations between microbiome and fear-related brain structure volumes. a** Scatterplot displaying relationship between 1-month Weighted Unifrac principal coordinate 1 and 1-year medial prefrontal cortex volume. Each dot represents a single subject. **b** Multiple linear regression model with *t* test of microbiome predictors at 1-month and 1-year medial prefrontal cortex brain volume, * denotes *p* < 0.05 but not meeting threshold for Bonferroni significance, *n* = 14 independent participants, covariates include age at scan and sex. **c** Scatterplot displaying relationship between 1-year Weighted Unifrac principal coordinate 1 and 1-year amygdala volume. Each dot represents a single subject. **d** Multiple linear regression model with *t* test of microbiome predictors at 1-year and 1-year amygdala brain volume, * denotes *p* < 0.05 but not meeting threshold for Bonferroni significance, *n* = 13 independent participants, covariates include age at scan and sex. Source data for this figure are provided as a Source Data file.

**Table 2 Predicted metagenomic functional ability associated with 1-month Weighted Unifrac principal coordinate 1.**

| KEGG L3 | q value[a] | p value[b] | estimate | std error | t | CI 2.5% | CI 97.5% | r² | df |
|---|---|---|---|---|---|---|---|---|---|
| Sphingolipid metabolism | 0.0017 | 1.22E−05 | −1.68E−06 | −1.68E−06 | −5.23 | −2.34E−06 | −1.02E−06 | 0.47 | 30 |
| Lysosome | 0.0017 | 7.89E−06 | −2.76E−06 | 5.12E−07 | −5.38 | −3.80E−06 | −1.71E−06 | 0.48 | 30 |
| Protein processing in endoplasmic reticulum | 0.0055 | 9.65E−05 | −6.98E−06 | 1.55E−06 | −4.49 | −1.02E−05 | −3.81E−06 | 0.40 | 30 |
| Protein digestion and absorption | 0.0055 | 7.20E−05 | −1.13E−05 | 2.45E−06 | −4.6 | −1.62E−05 | −6.26E−06 | 0.41 | 30 |
| Other glycan degradation | 0.0055 | 1.21E−04 | −9.91E−07 | 2.25E−07 | −4.414 | −1.45E−06 | −5.33E−07 | 0.39 | 30 |
| Glycosaminoglycan degradation | 0.0055 | 1.07E−04 | −3.34E−06 | 7.49E−07 | −4.46 | −4.87E−06 | −1.81E−06 | 0.39 | 30 |
| Meiosis yeast | 0.0138 | 3.56E−04 | 2.75E−05 | 6.84E−06 | 4.03 | 1.36E−05 | 4.15E−05 | 0.34 | 30 |
| Glycosphingolipid biosynthesis ganglio series | 0.017 | 5.12E−04 | −4.51E−06 | 1.16E−06 | −3.89 | −6.87E−06 | −2.14E−06 | 0.33 | 30 |
| Flavone and flavonol biosynthesis | 0.025 | 8.29E−04 | −2.49E−05 | 6.70E−06 | −3.72 | −3.86E−05 | −1.12E−05 | 0.31 | 30 |

Linear regression with t test, FDR correction for multiple comparisons.
[a]FDR correction for multiple comparisons.
[b]Linear regression with t test. Source data for this table are provided as a Source Data file.

fear reactivity and amygdala development, but could also be explained by the microbiome acting as a proxy for another variable. To better understand potential confounders, we tested for associations of medical, demographic, and feeding variables with microbiome measures of alpha diversity and beta diversity at 1 month and 1 year of age as well as 1-year behavioral outcomes. After FDR correction, maternal age at birth was positively associated with 1-month Weighted Unifrac PC2 ($n = 32$, $q = 0.001$, estimate = 0.018, Std. Error = 0.004, $t = 4.53$, CI = 0.01, 0.03, df = 30, $r^2 = 0.40$). There were no significant medical, demographic, or feeding variable associations with 1-year microbiome measures, IBQ-R fear index, Strange Situation response, or Mask Task outcomes. There were no significant relationships between 1-month Weighted Unifrac PC 2 and outcomes of behavior or brain volume to warrant sensitivity analysis with maternal age at birth as a covariate.

## Discussion

Experiments in animal models demonstrate the impact of the gut microbiome on fear behavior and its associated neurobiological substrates[24,59–61]. In this study, we observed that the human infant gut microbiome is also associated with differences in observed fear reactivity.

At 1 year, the beta diversity metric, Weighted Unifrac, was associated with fear behavior. Specifically, infants with negative Weighted Unifrac principal coordinate 1 values were more fearful in response to the Mask Task paradigm. Negative values on this principle coordinate indicate a greater relative abundance of *Veillonella*, *Dialister*, an unnamed genus of Clostridiales, *Bifidobacterium*, and *Lactobacillus*. This was further supported by a secondary analysis of individual genera which found that *Dialister* and an unnamed genus of Clostridiales were associated with increasing non-social fear, although these relationships did not survive FDR correction and should be considered non-significant trends. Infants with positive Weighted Unifrac PC 1 values were less fearful in response to the masks and had a greater relative abundance of *Bacteroides*. In keeping with our hypothesis, gut microbiome communities dominated by *Bacteroides* at 1 year were associated with less non-social fear. We also hypothesized that *Bifidobacterium* and *Lactobacillus* would be similarly associated with reduced fear, but the opposite was found. This discrepancy may be explained by the successional timing of gut microbiome development during infancy. The microbiome starts with a mix of aerotolerant bacteria like *Lactobacillus*, followed by transitions to facultative and obligate anaerobes[62–64]. Over the first year, oxygen concentration in the gut lumen declines significantly due to colonocyte metabolism of short chain fatty acids and luminal lipid oxidation[65,66]. This shapes the microbiome community by selecting for strict anaerobes like *Bacteroides*

which can better tolerate increasingly anoxic conditions. Consequently, infants on the positive side of PC1 may have a relatively more mature gut microbiome due to the lower abundance of *Lactobacillus* and increased abundance of *Bacteroides*. Fear reactivity can be reliably observed around 6 months of age, peaks at 12 months, and then decreases over the course of development. The relatively more mature 1-year gut microbiome may therefore act to accelerate the maturity of fear behavior as observed through lessened reactivity to non-social threat. The mechanism by which this occurs requires further elucidation through longitudinal investigations of the metabolic, immune, endocrine, and neural pathways of the microbiome-gut-brain axis. Regarding metabolic pathways, there were no significant associations of Weighted Unifrac PC1 with predicted metagenomic functional ability at 1 year of age. This may be due to the functional redundancy of the microbiome or due to the limitations of PICRUSt, including limited genomic reference and the inability to investigate differences in gene expression.

Whereas beta diversity examines differences in community composition between individuals, alpha metrics index the diversity of the gut microbiome within an individual. Principle component analysis of alpha diversity metrics at 1 month of age appears to separate measures of richness and phylogenetic diversity from a measure of evenness and richness (Shannon Diversity) as reflected by Alpha PC 2. Positive values on 1-month Alpha PC 2 are associated with larger Observed Species, Faith's Phylogenetic Diversity, and Chao1, but has a negative correlation with Shannon Diversity. Infants with positive Alpha PC 2 values (lower microbiome evenness, but not richness) at 1 month of age were more fearful in response to the masks at 1 year. This may suggest that the relative abundance (evenness) rather than simply the presence (richness) of particular taxa in the gut microbiome may be more important for developing fear behavior. One such mechanism would be through the gut microbiome metabolome. Taxa differentially produce various metabolic products including short chain fatty acids, polyphenols, amino acid derivatives, and neurotransmitters which have been shown to impact behavior[16,67,68]. Future metabolomics investigations of microbiome-associated metabolites would be equipped to further address this hypothesis.

Previous studies have shown that non-social and social fear reactivity in early development are not significantly correlated and may represent different constructs[56,69,70]. They also differ in their relation to psychopathology with high levels of social fear more strongly related to social anxiety disorders and high levels of non-social fear more strongly related to generalized anxiety and specific phobias[56]. However unusually low levels of either construct are associated with callous and unemotional traits[11]. Supporting previous research, fear reactivity observed in non-social and social fear paradigms was not correlated in this study. We

originally hypothesized that there would be associations of the gut microbiome with both types of fear, however we observed a significant relationship with non-social fear only. These findings can be considered in the context of the two-system theory of fear which posits that immediate and uncertain threats are processed through distinct neural circuits (amygdala vs bed nucleus of the stria terminalis)[71]. A stranger in the social fear paradigm may best represent an uncertain threat whereas the masks are novel, salient, and scary stimuli representing immediate threat. In addition, our findings parallel rodent literature where the impact of the microbiome on fear behavior has primarily been observed in non-social contexts like the elevated plus maze and open field assays[13,16,18]. The conserved relationship between the gut microbiome and defensive behavior in response to non-social threat across vertebrates may suggest a co-evolutionary relationship[72–75]. The microbiome's contribution to a host's behavioral response to immediate threat, like a predator, would result in survival benefits to both the host and microbiome.

Parent report of infant fear behavior at 1 year was not associated with microbiome community measures at 1 month or 1 year of age. The IBQ-R fear index encompasses parental report of fear behavior to novel situations in several non-social and social contexts. It has been previously shown to have low correlations with laboratory assessments of non-social and social fear reactivity[6]. The mix of questions regarding both non-social and social fear may explain the low correlation observed in our data. In addition, we detail significant findings with the Mask Task which is a highly arousing fear-evoking paradigm, in contrast to the IBQ-R fear index which focuses on novelty experienced in everyday life. Taken together, these reasons may explain why we do not see significant relationships with the same microbiome predictors associated with laboratory assessments of non-social fear.

Our results relating microbiome to non-social fear could reflect 3 different types of relationship (1) differences in microbial colonization cause differences in fear, (2) infants' fear responses influence colonization of the gut microbiome, or (3) the microbiome is serving as a proxy for some other factor. To address the latter explanation, we examined associations with a number of demographic, medical, and feeding variables. Compared to previous infant gut microbiome cohorts[54,57,63,76–82], we found very few significant associations of these variables with gut microbiome community measures. This may be due to the relatively small sample size or the strict inclusion/exclusion criteria used to recruit participants for this study. There may also be factors impacting gut microbiome community measures that were not measured in this study, such as the composition and microbiome of breast milk.

In addition to studying relationships between the infant microbiome and fear behavior, we also tested whether microbial composition was associated with gray matter volumes in three brain regions that are critical for fear behavior: the amygdalae, hippocampi, and prefrontal cortices. We found suggestive associations of the microbiome with medial prefrontal cortex volume and amygdala volume at 1 year of age, although these relationships do not survive Bonferroni correction. The same 1-year Weighted Unifrac PC 1 that was associated with non-social fear was also associated with amygdala volume. Infants with negative Weighted Unifrac PC 1 values had significantly more fear reactivity and also had larger amygdala volumes. Interestingly, a prior study of preterm infants found that larger newborn amygdala volume was associated with increased escape behavior in response to non-social fear at 1 year[83]. Larger basolateral amygdala volumes are also associated with increased levels of anxiety later in development (around 8 years of age)[84]. In our secondary analysis of genera, 1-month *Streptococcus* relative abundance had

a significant negative association with 1-month amygdala volume after FDR correction. Our results parallel preclinical literature demonstrating the impact of the microbiome on morphology of the amygdala[31] and clinical studies showing that functional connectivity of the amygdala is associated with the microbiome at 1 year of age[85]. Future studies using functional connectivity or diffusion tensor imaging to assess connectivity between these brain regions involved in fear may yield more insight.

In conclusion, this study is an important step in demonstrating the relevance and importance of the microbiome in human neurodevelopment. Our results suggest that the infant gut microbiome may contribute to the developmental trajectory of fear reactivity and that this relationship may involve the amygdala. Behavioral inhibition in infancy, as measured in this study, predicts future internalizing psychopathology as an adult[10]. As such, this work may have implications for psychiatric disorders and behavioral problems characterized by abnormal fear reactivity including social anxiety, phobias, or callous-unemotional traits. Strengths of this study include the longitudinal prospective design, reduction of potential confounding variables through the selection of a healthy, breastfed, vaginally delivered cohort, focus on development, observational assessment of fear reactivity, use of magnetic resonance imaging, and conservative statistical approach. The study also had certain limitations. First, as a pilot study, the sample size was relatively small by design. Small sample sizes can produce unstable results and we acknowledge there is the risk of homogenous sampling leading to statistical inferences that do not represent the overall population. Despite this, we did observe several significant associations with conservative Bonferroni and false discovery rate thresholds which supported our a priori hypotheses. Expected effect sizes are currently unclear due to the early discovery nature of human microbiome-gut-brain axis research. As such, these findings should be treated with caution until replicated. Second, the unique inclusion and exclusion criteria of this study may limit generalizability. Futures studies of larger and more diverse cohorts could address this and also potentially allow for examination of more subjects on the extremes of fear reactivity who are at the most risk for later psychopathology. Third, this study did not attempt to identify causal mediators between the gut microbiome and brain development or behavior. Future investigations should probe microbiome-gut-brain axis measures including gut metabolites, cortisol reactivity, vagal signaling, and immune system programing[86]. Fourth, we purposefully did not assign positive or negative valence to the microbiome and behavioral outcomes described in this study. Given that research on the infant microbiome in relation to behavioral and health outcomes is still in the discovery phase, we felt it would be premature to label certain microbiome metrics as "good" or "bad." Indeed, it may be the case that microbiomes that are good for one outcome are bad for other outcomes. In regard to fear behavior, we know that extremes of fear behavior at 1 year of age are linked to future anxiety-disorders (high reactivity) and callous-unemotional traits (low reactivity), but fear behavior itself is a part of normal development. Rather than assign pathologic significance, we detailed associations of the human infant microbiome with variation in fear behavior as previously described in preclinical animal models. Fifth, while 16S rRNA amplicon sequencing is often utilized by gut microbiome studies, it cannot resolve species or strain differences and is limited to the inference of functional genomic capability (PICRUSt). Future studies utilizing shotgun whole genome sequencing or transcriptomics would provide analysis of the functional capability of the microbiome beyond identification and relative quantification.

While the associations detailed in this study do not prove causality, they provide impetus for future investigations into the

### Table 3 Data collection by visit.

| | Visit 1 (1 month) | Phone Interview (6 months) | Visit 2 (1 year) |
|---|---|---|---|
| Sociodemographic | × | | × |
| Medical[a] | × | × | × |
| Feeding History[b] | × | × | × |
| State Trait Anxiety Inventory | | | × |
| Life Experiences Survey | | | × |
| Fecal Sample | × | | × |
| Infant Behavior Questionnaire-Revised | | | × |
| Neuroimaging | × | | × |
| Strange Situation | | | × |
| Lab-TAB Mask Task | | | × |

[a]Medical history from parent report and medical record review.
[b]Adapted from CDC Infant Feeding Practices Survey II[106] and NHANES Child Feeding Questionnaire.

complex interplay between the gut microbiome and brain as well as microbiome predictors of later neuropsychiatric pathology. With further research, the gut microbiome may emerge as a key modulator of fear development and as such may become a means to prevent or ameliorate psychiatric disorders and behavioral problems characterized by abnormal fear reactivity.

## Methods

**Study population and study visits.** We recruited 34 infants from UNC and REX Hospitals in central North Carolina as participants in this prospective longitudinal cohort pilot study. Inclusion criteria for participation in the study were vaginal delivery and exclusive breastfeeding until the first study visit at 1 month. Participants were excluded for maternal antibiotic usage two weeks before delivery (including Group B Streptococcal prophylaxis), antibiotics given to the infant before the first study visit, neonatal intensive care unit stay, birth weight <2500 g, gestational age <37 weeks, major maternal medical illness, prenatal drug use, primary language other than English, and fetal ultrasound abnormalities. Prenatal, labor and delivery, and pediatric medical records were reviewed to ensure that participants met study inclusion/exclusion criteria. Participants had two study visits at UNC (median age at visit 1 = 30 days, range 15–59 days; median age at visit 2 = 384 days, range 333–491 days) and one phone interview at 6 months. Data collected at each visit are displayed in Table 3 and include sociodemographic, medical, feeding history, State Trait Anxiety Inventory[87], Life Experiences Survey[88], fecal sample, Infant Behavior Questionnaire-Revised[89], neuroimaging, Strange Situation[90], and Lab-TAB Mask Task[91].

See Table 4 for cohort description (sample size may differ between variables due to non-response, or loss to follow-up at 6mo or 1 yr) (binary variables were included only when the least common response was given >20% of the time). Informed written consent was obtained from parent/legal guardian of each participant. This study was approved by the Institutional Review Board of the University of North Carolina at Chapel Hill.

**Microbiome analysis—DNA isolation.** Participating families were mailed a sample collection kit shortly before each visit that included 2 tubes (one for backup) each containing 1 ml Allprotect reagent (Valencia, CA). Parents were instructed to collect ~200 mg of feces from a single soiled diaper, immediately place it in a tube completely submerged in reagent, and bring to the study visit (samples submerged in Allprotect can be stored up to 7 days at 15–25 °C). Once received, the tubes were stored at −80 °C until analysis. All microbiome analysis including DNA isolation, sequencing, and sequencing data analysis was completed in separate batches for 1-month and 1-year samples. DNA isolation was performed as described in[92,93]. Specifically a portion of the collected stool samples (~200 mg) were combined with 200 mg of 212–300 μm glass beads (Sigma, St. Louis, MO) and 1.4 ml of Qiagen ASL buffer (Valencia, CA) in sterile 2 2 ml tubes. Bead beating commenced for 5 min in 1-min intervals in a Qiagen TissueLyser II at 30 Hz. Next, samples were incubated at 95 °C for 5 min and centrifuged at $21000 \times g$ for 5 min. To remove PCR inhibitors, supernatants were transferred to new 2 ml-tubes containing InhibiEx inhibitor adsorption tablets (Qiagen) and vortexed vigorously. After a brief centrifugation, supernatants were aspirated and combined with Qiagen AL buffer and Proteinase K (600IU/μl). After a 10-min incubation at 70 °C, DNA was purified via a standard on-column method using Qiagen buffers AW1 and AW2 as washing agents and eluted in 10 mM Tris (pH 8.0).

**16S rRNA amplicon sequencing.** Sequencing of the generated amplicons targeting the V1-V2 region of the bacterial 16S rRNA gene[94–96] plus *Bifidobacterium*-specific primers in a 4:1 Universal to *Bifidobacterium* was carried out on the Illumina MiSeq platform as described in[57,80]. The complete sequences of the primers are available in Supplementary Table 5. Master mixes for PCR amplification contained 12.5 ng of total DNA, 0.2 μM of each primer and 2x KAPA HiFi HotStart ReadyMix (KAPA Biosystems, Wilmington, MA). Amplification commenced with the following steps: 95 °C for 3 min, cycling of denaturing of 95 °C for 30 s, annealing at 55 °C for 30 s and a 30 s extension at 72 °C (25 cycles), a 5-min extension at 72 °C, and a final hold at 4 °C. The generated amplicons were purified using AMPure XP (Beckman Coulter, Indianapolis, IN) and samples were next amplified using a limited cycle PCR program, adding Illumina sequencing adapters and dual-index barcodes (index 1(i7) and index 2(i5)) (Illumina, San Diego, CA) to the amplicon target. Amplification of each sample commenced with the following steps: initial denaturing at 95 °C for 3 min, followed by a denaturing cycle of 95 °C for 30 s, annealing at 55 °C for 30 s and a 30 s extension at 72 °C (8 cycles), a 5-min extension at 72 °C and final hold at 4 °C. The final libraries were again purified using the AMPure XP reagent, quantified and normalized prior to pooling. The DNA library pool was then denatured with NaOH, diluted with hybridization buffer and heat denatured before loading on the MiSeq instrument (Illumina). Automated cluster generation and paired–end sequencing with dual reads were performed according to the manufacturer's instructions.

**Sequencing data analysis.** After the sequencing run, BclToFastq (Illumina) was used to produce multiplexed paired-end fastq files, which were joined into a single multiplexed, single-end fastq file using fastq-join as described in[57,80]. After demultiplexing and quality filtering, quality analysis reports were generated with FastQC. Total reads per sample ranged from $5.18 \times 10^3$ to $3.15 \times 10^5$ at 1 month and $1.38 \times 10^4$ to $1.47 \times 10^5$ at 1 year. One sample was excluded at 1 year for reads <0.1% of total. Bioinformatics analysis of bacterial 16S rRNA amplicon sequencing data was conducted using the Quantitative Insights Into Microbial Ecology (QIIME) software[96]. OTU picking was performed on the quality filtered results using pick_de_novo_otus.py. Chimeric sequences were detected and removed using ChimeraSlayer. Summary reports of taxonomic assignment by sample and all categories were produced using QIIME summarize_taxa_through_plots.py and summarize_otu_by_cat.py. Alpha and beta diversity analysis were performed on the data set using the QIIME routines: alpha_rarefaction.py and beta_diversity_-through_plots.py[97,98], respectively. Rarefaction was set at 5000 for each timepoint. Alpha diversity is a measure of within-individual diversity while beta diversity measures dissimilarity between individuals. Alpha diversity measures include Shannon Diversity (measure of richness and evenness), Observed Species (richness), Faith's Phylogenetic Diversity (phylogenetic measure expressed as tree units observed), and Chao1 (estimate of total OTUs that would be observed with infinite sampling). Principal component analysis of the 4 alpha diversity measures at each age were used as the main predictors in subsequent analyses. Alpha PC 1 and Alpha PC2 explained 78 and 13% of the variance at 1 month and 90 and 5% of the variance at 1 year, respectively. Weighted Unifrac is a beta diversity measure that incorporates the relative abundance of taxa. Principal coordinates analysis of Weighted Unifrac was run separately at each timepoint. Weighted Unifrac PC 1 and 2 explained 53 and 12% of the variance at 1 month and 68 and 13% of the variance at 1 year, respectively. In total, useable microbiome data was obtained for 32 and 21 participants at 1 month and 1 year of age, respectively.

**Prediction of metagenome functional content.** Phylogenetic Investigation of Communities by Reconstruction of Unobserved States (PICRUSt) was used to predict metagenome functional content from the 16S rRNA sequencing data[58]. The PICRUSt pipeline was used to predict the abundance of Kyoto Encyclopedia of Genes and Genomes (KEGG) orthologs by participant at the collapsed KEGG pathway levels.

**Predictor and outcome covariate identification.** Relevant variables were created based on literature review of important factors that may influence early gut microbiome development or performance on measures of fear behavior. We used linear models to identify environmental variables that could act as confounders due to their association with beta diversity, alpha diversity, or behavioral outcomes. Binary categorical variables were filtered by requiring >20% frequency in the study population. The p values were combined and corrected for multiple comparisons using FDR. Variables with q values less than 0.05 were considered significant. See Supplementary Table 6 for list of variables assessed and how they were created.

**Behavioral assessments.** The Strange Situation is a well-established paradigm designed to assess infant attachment as the caregiver and a stranger enter and exit the room[90]. For this study, we analyzed Episode 3 of the Strange Situation to assess wariness or social anxiety of the infants at 1 year of age. In this 3-min episode, the infant played alone with toys on a blanket while their mother read a magazine. For the first minute, a male stranger entered the room and read quietly. During the second minute, the stranger engaged the mother in conversation and gradually directed more attention toward the infant. In the final minute, the stranger sat on

**Table 4 Cohort description.**

**Categorical variables**

| Descriptive variable | N | % | Descriptive variable | N | % |
|---|---|---|---|---|---|
| Sex | | | Formula Feeding at 6 Mo. | | |
| Male | 23 | 67.6 | Yes | 15 | 46.9 |
| Female | 11 | 32.4 | No | 17 | 53.1 |
| Income | | | Cereals at 6 Mo. | | |
| High | 13 | 38.2 | Yes | 20 | 74.1 |
| Middle | 13 | 38.2 | No | 7 | 25.9 |
| Low | 8 | 23.5 | Breastfeeding at 1 Year | | |
| Maternal race | | | Yes | 15 | 50.0 |
| White | 26 | 76.5 | No | 15 | 50.0 |
| Black | 8 | 23.5 | Formula Feeding at 1 Year | | |
| *Paternal Race* | | | Yes | 13 | 43.3 |
| White | 27 | 79.4 | No | 17 | 56.7 |
| Black | 7 | 20.6 | Nuts at 1 Year | | |
| Maternal Pre-Pregnancy BMI | | | Yes | 20 | 66.7 |
| Under | 1 | 2.9 | No | 10 | 33.3 |
| Normal | 25 | 73.5 | Sweet Foods/Drinks at 1 Year | | |
| Overweight | 6 | 17.6 | Yes | 21 | 70.0 |
| Obese | 2 | 5.9 | No | 9 | 30.0 |
| Maternal Infection During Pregnancy | | | French Fries at 1 Year | | |
| Yes | 11 | 32.4 | Yes | 18 | 60.0 |
| No | 23 | 67.6 | No | 12 | 40.0 |
| Maternal Psychiatric History | | | Milks at 1 Year | | |
| Yes | 9 | 26.5 | Yes | 23 | 76.7 |
| No | 25 | 73.5 | No | 7 | 23.3 |
| Older Siblings | | | Antibiotic Usage in First Year | | |
| Yes | 20 | 58.8 | Yes | 13 | 43.3 |
| No | 14 | 41.2 | No | 17 | 56.7 |
| Vitamin D at 1 Mo. | | | Daycare Attendance | | |
| Yes | 13 | 38.2 | Yes | 9 | 34.6 |
| No | 21 | 61.8 | No | 17 | 65.4 |
| Vitamin D at 6 Mo. | | | Fever in Last 2 Weeks at 1 Year | | |
| Yes | 8 | 28.6 | Yes | 8 | 26.7 |
| No | 20 | 71.4 | No | 22 | 73.3 |
| Continuous Variables | | | | | |
| Descriptive Variable | N | Mean ± SD | Descriptive Variable | N | Mean ± SD |
| Gestational Age at Birth (days) | 34 | 275.5 ± 7.6 | Age at Visit 2 (days) | 31 | 387 ± 37.6 |
| Birth Weight (grams) | 34 | 3330 ± 397 | Month Food Introduced[a] | 32 | 5.6 ± 1.9 |
| Maternal Age at Birth (years) | 34 | 30.3 ± 4.8 | Maternal State Anxiety[b] | 27 | 32.3 ± 10.9 |
| Maternal Education (years) | 34 | 16 ± 2.2 | Maternal Trait Anxiety[b] | 27 | 31.9 ± 8.3 |
| Paternal Age at Birth (years) | 34 | 32.6 ± 6.9 | Negative Life Events[c] | 27 | 3.6 ± 5.4 |
| Paternal Education (years) | 34 | 15.8 ± 2.8 | Positive Life Events[c] | 27 | 5.4 ± 5.2 |
| Age at Visit 1 (days) | 32 | 30.2 ± 10.8 | Total Life Events[c] | 27 | 9.0 ± 6.6 |

All socioeconomic characteristics are based on maternal report.
[a]Introduction of food other than breastmilk or formula.
[b]State-Trait Anxiety Inventory for Adults[87].
[c]Life Experiences Survey[88].

the floor and attempted to engage the infant with toys. Video cameras captured wide angle views of the room for later coding.

To assess the expression of non-social fear, we adapted the Masks portion of the locomotor Laboratory Temperament Assessment Battery[99]. In this assessment, a research assistant wore a series of masks (apple, horse, monkey, alien) and presented each individually to the infant seated in a high chair. Each mask was presented for ~10 s while the research assistant said the infant's name 3 times. Video cameras recorded the infant's reaction to the masks for later coding. Research assistants conducting behavioral assessments were blind to microbiome outcomes. The Mask Task trial was ended early if the participant demonstrated high distress (continuous hard crying for more than 10 s). In addition, if the infant became upset by the high chair before the start of the Mask Task trial, the paradigm was not started as it would confound interpretation ($n = 10$).

Before visit 2, mothers were mailed a copy of the Infant Behavior Questionnaire – Revised (IBQ-R)[89] to complete and bring to the study visit. Questionnaires were scored to generate a composite fear score to use as a parent reported infant fear behavior outcome at age 1 year.

**Behavioral assessment coding.** Strange Situation Episode 3 was divided into three sections (3.1, 3.2, 3.3) and was coded on a global 3-point scale for social

wariness toward the stranger during each section. A score of 1 indicates little to no wariness where the infant explores the space, plays with toys, and does not cry, freeze, or cling to the caregiver or caregiver's chair for more than 5 s. A score of 2 indicates minor to moderate wariness where the infant may demonstrate freezing behavior or clinging to caregiver or caregiver's chair for more than 5 s but less than 30 s. A score of 3 indicates moderate to marked wariness where the infant demonstrates any crying, persistent clinging to caregiver or caregiver's chair for more than 30 s or freezes for more than 30 s.

Details for coding of the Mask Task assessment are found in Supplementary Table 7. Briefly, fear in response to each mask was coded for facial fear, vocal distress, bodily fear, escape behavior, and startle response on a 0–3 scale for increasing intensity of fear behavior. No Mask Task data warranted exclusion for interfering parent behavior. All videos for Strange Situation and Mask Task were coded by one person who was reliable with a second coder with >0.8 ICC. Both coders were blinded to microbiome predictors.

**Image acquisition.** Scans were acquired on a Siemens 3 T TIM-Trio scanner (Siemens Medical System, Erlangen, Germany) during unsedated natural sleep at both study visits. Magnetization prepared rapid acquisition gradient-echo (MP RAGE) scans and T2 weighted scans were obtained at visit 1 with the following

parameters: MP RAGE (repetition time = 1900 ms, echo time = 3.89 ms, 7 degree flip angle, $0.8 \times 0.8 \times 0.8$ mm voxel resolution), 3D T2 weighted (turbo-spin echo sequence, repetition time = 3200 ms, echo time = 406–410 ms, 120 degree flip angle, $0.8 \times 0.8 \times 0.8$ mm voxel resolution). MP RAGE scans were obtained at visit 2 with the following parameters: (repetition time = 1900 ms, echo time = 3.1 ms, 7 degree flip angle, $0.8 \times 0.8 \times 0.8$ mm voxel resolution). A total of 3 infants at visit 1 and 10 infants at visit 2 did not go to sleep or woke up in the scanner (success rate of 91 and 62% respectively).

**Structural image analysis**. Images were examined for quality control to exclude scans with motion or imaging artifacts. Two scans failed visual quality control for motion/artifact at 1 month, with no scans failing visual quality control at 1 year of age. The structural processing consisted of brain tissue classification into gray matter, white matter, and cerebrospinal fluid. An age specific atlas based expectation maximization segmentation algorithm was employed, operating jointly on T1- & T2-weighted images for visit 1, and only on T1-weighted images at visit 2 utilizing software AutoSeg 3.3.2, NeoSeg 1.0.8, and ITK-SNAP 3.4[100]. Gray matter was subdivided into 83 regions via a multi-atlas based fusion procedure[101] of the Gousias pediatric template database[102]. In order to ensure a consistent parcellation in both neonate and 1 year old setting, we applied this 1 year pediatric template to all datasets independent of age. Finally, all parcellation results were visually assessed for appropriate parcellation performance. A single scan at 1 month of age failed segmentation QC, resulting in 29 participants with useable brain volumes. No scans at 1 year failed segmentation QC resulting in 16 participants with useable brain volumes. Right and left hippocampus and amygdala gray matter volumes were summed into a single bilateral volume measurement. Total medial prefrontal cortex gray matter volume was generated through the sum of bilateral cingulate gyrus anterior, straight gyrus rectus, superior frontal gyrus, medial orbital gyrus, subgenual frontal cortex, and pre-subgenual frontal cortex.

**Statistical methods**
*Description of statistical models*. Linear mixed effect models with random intercept or multiple linear regression models (R 3.5.1) were used to test for effects of alpha and beta diversity on non-social fear behavior, social fear behavior, IBQ-R fear index, and brain volumes. Alpha and beta diversity metrics, Alpha PC 1 & 2 and Weighted Unifrac PC 1 & 2, were all included as multiple predictors in these models. Models with Strange Situation fear outcomes from episodes 3.1, 3.2, and 3.3 used a one-level mixed effect model to account for the within-subject covariance of successive episodes. For the Mask Task with both multiple correlated outcomes (facial fear, vocal distress, bodily fear, startle, escape behavior) and multiple episodes (up to 4 different masks presented), we adapted a two-level mixed effects structure to account for the within-subject correlations among different fear outcomes and within-subject but between-mask correlations of the same outcome. Satterthwaite's methods was applied to approximate degrees-of-freedom[103]. Multiple linear regression models were used for outcomes involving IBQ-R fear index or brain volume outcomes as well as analyses to identify potential covariates from medical, demographic, and feeding variables associated with the infant gut microbiome. Any identified variables were included as covariates in subsequent sensitivity analyses of relevant brain and behavior testing. Models with brain volume outcomes included age at scan and sex as covariates.

*Genera analysis*. For the secondary analysis of genera associated with non-social fear behavior and brain volumes, bacterial genera were first selected for analysis by the following criteria in order to remove rare outliers and select for genera dominant in microbiome composition: (1) more than 20% of the subjects had non-zero relative abundance for that genus and (2) the 90% quantile of relative abundance for that genus within the cohort was larger than 0.5%[104].

*Multiple comparison correction*. For primary analyses (associations between the microbiome and fear reactivity), we used a Bonferroni correction which takes account of both the number of predictors (4 microbiome measures—Weighted Unifrac PC 1 & 2, Alpha PC 1 &2) and the number of models that were run (2 models for the Mask Task, 2 for the Strange Situation paradigm, and 2 for the IBQ-R, with one model examining associations with the 1-month microbiome and one model examining associations with the 1-year microbiome). For secondary analyses (associations between the microbiome and specific brain volumes), we also used a Bonferroni correction which takes account of both the number of predictors (4 microbiome measures) and the number of models that were run (1 model for associations between the 1-month microbiome and 1 month brain volumes, 1 model for associations between the 1-month microbiome and the 1-year brain volumes, and 1 model for associations between the 1-year microbiome and the 1-year brain volumes). Exploratory analyses testing for associations between individual genera and non-social fear reactivity were adjusted using false discovery rate correction, as were our exploratory analyses testing for associations between individual genera and brain volumes.

**Reporting summary**. Further information on research design is available in the Nature Research Reporting Summary linked to this article.

## Data availability
16S rRNA amplicon sequencing data is available through the NCBI repository under accession PRJNA547558. The KEGG database used in PICRUSt analyses is accessible at https://www.genome.jp/kegg/. Some of the data collected for this project is identifiable and/or sensitive and for this reason is only available upon request. Individuals interested in obtaining this data should contact the corresponding author. Sharing of this data may be possible, subject to IRB review and the execution of an appropriate data use agreement. Source data are provided with this paper.

## Code availability
Code used to conduct the analyses described in this paper is available at https://github.com/argossy/gmia_public and was also deposited at Zenodo[105].

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

## Acknowledgements

Jennifer Quesenberry, Wendy Neuheimer, and Dianne Evans served as study coordinators. Nguyen Yu and Sierra Atwater assisted with behavioral assessments. Corey Shope, DeSiyre Spurgeon, and Preman Koshar provided study administrative assistance. Joe Blocher at the Neuro Image Research and Analysis Laboratories provided assistance with image analysis. We thank the families for their participation in this study. This study was supported by NIH Grants T32 NS007432 and T32 GM008719 to A.L.C., P30 DK34987 to M.A.A., NARSAD to K.X., T32 MH106440 to J.P.F. and S.P.R., R21 MH104330 and R33 MH104330 to R.C.K., and the NSF (GRFP Fellowship DGE-1650116 to S.P.R.).

## Author contributions

Conceptualization, R.C.K.; Statistical Analysis, K.X., M.A.A., S.R., J.P.F., W.M.; Image analysis, M.A.S., A.L.C., J.B.Z.; Behavior assessment and analysis: A.L.C., J.B.Z., M.C.K. C.B.P.; Microbiome analysis, M.A.A.; Data Curation, A.L.C.; Writing—Original Draft, A.L.C.; Writing—Review & Editing, A.L.C., R.C.K., K.X., C.B.P., M.A.A., S.R., J.P.F., J.B.Z., M.C.K., M.A.S., A.L.T.; Visualization, A.L.C. and K.X.; Supervision, R.C.K.; Project Administration, A.L.C., and R.C.K.; Funding Acquisition, R.C.K.

## Competing interests

R.C.K. is a consultant on a grant from Nestle/Wyeth entitled "Interrelationships of Nutrition, Gut Microbiota, as well as Brain and Cognitive Development in Early Life." The other authors report no financial interests (biomedical or non-biomedical) or potential conflicts of interest.
