## [Peer Review File · Nature Communications]

Reviewers' comments:

Reviewer #1 (Remarks to the Author):

In this study, the authors present novel data showing associations of the human infant gut microbiome with non-social fear behavior and amygdala volume.

Although the sample size is relatively small, the study represents an important step forward in translating animal data into humans.

Given the multiple testing that were carried out it will be important to clarify whether the p values were adjusted for multiple comparisons.

Reviewer #2 (Remarks to the Author):

This is a fascinating and important investigation of links between the gut microbiota and fear behavior and amygdala volume in infants. There are many strengths of the study, including its longitudinal design, the methods chosen, and the analytic procedures. There are also some significant weaknesses of the study that require further consideration and elaboration.

First, the sample size is a major limitation, and although it is acknowledged by the authors, it gives me pause when examining behavioral measures that include as few as 14 infants. Although the behavioral measures used are among the best that are available for inferring fear behavior in infants, they are nevertheless complicated by large measurement error and basic difficulties, such as getting infants to sit in a highchair and attending to the task without crying (as experienced by the authors in this investigation). These difficulties are commonly experienced in this kind of work. I appreciate the careful exclusion of such infants in the current report along with the attention to interrater reliability. All these considerations are essential, but they do not compensate for the small sample size. Measuring behavior in infants is challenge and messy work and this is a major reason why a larger sample size is required to have confidence in the findings. Small sample sizes do not only suffer from type 1 errors, they are also prone to type 2 errors that arise for many reasons, including chance associations based on small error that can sometimes occur in small sample size. I also appreciate that the authors acknowledge the need for replication, but this is a requirement for all science, and the issue at hand is whether one can have confidence that the associations described here are in fact real. I also appreciate the framing of this study in terms of pilot work, nevertheless the focus is on hypothesis testing and p-values which means that it must be evaluated on this basis and not on the basis of effect sizes or other considerations that might be important to pilot work. My confidence in the conclusions would be increased with a larger sample size.

Second, some of my initial concerns about sample size were assuaged by the consistent negative association between the WU PC1 and amygdala volume at both 1-month and 1-year. On the surface at least, this consistency suggests reliable underlying processes. However, analysis of the change in microbiome from 1-month to 1-year reveals that the WU PC1 component is actually very different at 1-month versus 1-year. Specifically, at 1-month, WU PC1 has a positive association with Bifidobacterium and negative association with Bacteroides but at 1-year the pattern is exactly opposite. These 2 genera are obviously only 2 among many, so I don't want to extend the point too far, but from figure 4 we can see that at 1-month Bacteroides has the most negative loading of any genera on WU PC1 whereas at 1-year it has the strongest positive loading on the same component. From a developmental perspective the change in composition of the microbiome may make sense, as described by the authors, but what requires further consideration is how the WU PC1 at these different timepoints with very different loading of genera like Bacteroides can have a consistently negative association with amygdala volume. I suspect that the answer to this puzzle lies in the developmental change that is described.

Third, the developmental change in the microbiome is not well integrated into the paper. It is described neither in the hypotheses nor the discussion. Furthermore, the justification for the 1-month and 1-year timepoints is lacking. Certainly, this kind of analysis may help to elucidate potentially confusing patterns of association as I describe above, but beyond that, such analysis

could provide useful insight into developmental processes that may contribute to fear behavior in children. I find the developmental change described in figure 5 fascinating and potentially important to the focus of this paper, but this potential has not been realized.

Fourth, I have a concern that some of the interpretation is not well justified by the data. For example, Fig 4 shows the taxa associated with each PC, but no data show whether these taxa are correlated with fear behavior or amygdala volume. The associations described are indirect: fear and amygdala to the PCs and PCs to taxa. However, the authors still made a connection between these taxa with fear behavior based on it. At minimum, this disconnect needs to be acknowledged. Additional minor points.

Line 59. The stimuli are not what emerges, but rather the response to them.

Line 61. Infants begin processing fear prior to 6-months of age, however their behavioral responses become reliable and strong enough by 6-months to allow their reliable measurement.

Line 73. Please indicate the direction of the effect.

Line 76-77. As a general principle, the idea that timing of exposure to the microbiome is important to the acquisition of fear behavior does not make sense. The statement is certainly true of germ free mice. But all humans are exposed to microbiota from birth, and perhaps prior, thus the timing of exposure in humans means something rather different from GF mice.

Line 91, please indicate the direction of the effect.

Line 121: The author mentioned that to reduce the confounding effect, one inclusion criteria is that all infants were breastfed until 1-month old. However, the influence of feeding practice to the gut microbiota can last until weaning age or even 1-year old. How is the feeding practice effect being adjusted for participants at 1-year age?

Line 161. It would be useful to add '1 year of age' to the title for Table 1. I found myself a bit confused by what was measured at what timepoints, and this might help.

Line 194. Given that the effect in figure 2 is not reliable, because it is known that it is driven by a single individual, it should be eliminated from the paper. Ensuring that the data is appropriate for analysis is a step prior to analysis, and findings with uncleaned data should not be reported. I do understand in this context that such findings may be intriguing because of the exploratory nature and because of the importance of not missing associations that may be present when the sample size is small, but in this case the effect is known to not be reliable because of the outlier and therefore should be reported as such.

Line 286. Thank you for including Table 3, which provides important 'background' information. It does not relate to the hypotheses and should be moved to the supplemental material.

Line 291. The inclusion of covariates is important, and the included sensitivity analysis is important but not discussed. One is left wondering if vitamin D and maternal age really have anything to do with the associations or whether they are there by chance. It would be very useful to discuss the biological plausibility of these covariates in relation to the associations under investigation.

Lines 352 – 356. I am not clear on how studies on asthma and impairment of the immune system inform the suggestion on line 358ff. They don't seem to follow.

Line 425. I do agree on the strengths of the study, but the selection of subjects also should be addressed as a limitation to the generalizability and interpretation of the data.

Line 631. Please include a description of the statistical model that was used. My interpretation is that a linear mixed model was used in which the repeated measures of fear behavior was used as an outcome, for each fear measure separately. This makes sense to me for the behavioral tasks because it can be broken down into multiple epochs but does not make sense to me regarding the IBQ R which only has one value. Please clarify. Also, please indicate which parameters were estimated in these models.

Line 768. One of your key citations is incomplete, which prevents evaluating an important claim in the paper.

Reviewer #3 (Remarks to the Author):

I wish to thank both the authors and the editors for the opportunity to review this manuscript. The authors are to be commended for their innovation in asking some very interesting questions, and the rigor of their behavioral data collection and coding.

In essence, the investigators use two measures (alpha diversity and taxonomic community

structure) to link the infant gut microbiome development with behavior in the first year of life. They observe significant coarse correlations, which are both interesting and novel for human cohort studies. However, there are also several major concerns.

1. There is an inverse relationship between alpha diversity and social fear, which held true at several relative abundance measures of beta diversity but did not hold true in inferred metagenomics analyses.

I find this troubling for two reasons. First, decreased alpha diversity is most related to breast milk and specifically mothers own milk feeding. The investigators seem to suggest that this diminished alpha diversity is a "bad thing" ergo lending to social fears. However, this could simply be a pathway to adjustment.

It would be helpful to more formally integrate with GLM models (or proc model) the mode and manner of feeding with the fear, and then do a multipartite analysis with both alpha and beta diversity.

Second, I find it concerning that there is no functional metagenomics relationship. While this may be due to the coarseness and inference approach they used (PICRUST), it is crucially important to discuss and attempt to further understand. In other words, if the pathways don't actually differ, are they only measuring small but non-relevant community membership differences?

2. The initial correlation matrices don't seem to ever be fully resolved in the multidimensional analyses. Perhaps I am missing some key aspects to their analysis, but it would appear that (as anticipated) there is significant interaction among behaviors. This is certainly expected. However, it is less clear how and if they looked for these same interactions as explanatory of these rather small microbiome alpha diversity measures.

3. A real lost opportunity with the study in its current form is the absence of MDD or Dirilecht modeling over time (i.e., longitudinal measures of community composition and change). In other words, is there a difference in the stability or dynamic nature of the maturation of the communities that correlates/responds with fear responses? There are several good ways to model and measure with the gut microbiome, and this would be important to look at.

Reviewer #4 (Remarks to the Author):

Summary

This study investigates the gut microbiome composition and diversity at 1 month and 1 year of age, and associates these with fear-related behavior and volumes of amygdala, hippocampus and medial prefrontal cortex, also measured at 1 month and 1 year of age. In the 1-month associations there is around $n=19$ for the behavioral associations and $n=27$ for brain volume associations, and at 1 year, there are around $n=14$ for the behavioral associations and $n=13$ for the brain volume associations. There were negative associations between microbiome diversity and fear at 1 month, and with fear and microbiome composition at 1 year. There was a negative association between 1 month (and 1 year) microbiome diversity and 1 month amygdala volume, but positive association with 1 year amygdala volume. The authors conclude that they are first to demonstrate associations of gut microbiome with fear behavior and the amygdala (a fear-related brain structure).

This is a generally well-written manuscript with a novel and interesting topic. My main concern is that the numbers used in this study are very small, and therefore we cannot be sure that the effects reported are real. The associations and their direction seem a bit random, and may be due to multiple comparisons, outliers, or confounders that I'm not sure are properly dealt with.

Abstract

An idea of the numbers of data for each analysis at each timepoint would be helpful to the reader.

Introduction

The introduction is excellent and motivates the study very well.

Results

It is unclear whether the results reported in figure 1-3 are corrected for multiple comparisons?

There are many instances where the direction of the results is not clearly stated.

The number of abbreviations makes it difficult for the reader to follow.

Has KEGG abbreviation been defined?

Again, the direction of the associations is sometimes not stated in the 'identification of covariates' section.

I would have thought you would add both identified covariates to the sensitivity analyses together?

I am concerned that some of the covariates that were not found to be 'significantly associated' with the microbiome or behavioral measures may still be important (especially since multiple comparisons correction was applied here, but not for the main analysis, if I understand correctly?)

Potential influencers of brain volume are not considered for those analyses, it is well known that sex and age at scan affects brain volume, for instance.

It is fine that you chose particular brain regions hypothesized to be involved in fear behavior, but I also wonder if you would find random other associations with other brain regions, i.e. would these brain regions be uniquely and specifically associated with the microbiome?

Discussion

It was difficult to get an idea of the main interpretation of the findings from the discussion. I found it hard to understand which direction of the microbiome metrics were 'good' or 'bad' for instance, and likewise for the species composition.

The discussion of results seems a bit selective, based on how the narrative is framed by the authors. There is not much discussion of the results that remained from the sensitivity analyses, for instance. Also, line 403-410 doesn't discuss the 1-month WU associations (e.g. with 1-year amygdala, which was the only one significant after covariate adjustment).

I'm not sure of the relevance of some of the inflammation discussion regarding linking microbes and anxiety.

The conclusion is good in that it does not overstate these findings.

Methods

It is good that the inclusion criteria removed some of the potential confounders at 1 month, but one wonders about their confounding effects later. It is difficult to deal with these properly, since the sample size is so small.

Please state the range of age at each visit.

Typo 'at' on line 548.

The sequencing details seem comprehensive, but I am not qualified to comment on this methodology!

Is the Gousias template for 2-year-olds? It might be worth discussing potential issues with using this on 1 month and 1-year MRI data.

The statistical methods section is a bit sparse, and in particular does not mention correction for multiple comparisons.

Reviewer #1 (Remarks to the Author):

In this study, the authors present novel data showing associations of the human infant gut microbiome with non-social fear behavior and amygdala volume.

Although the sample size is relatively small, the study represents an important step forward in translating animal data into humans.

Given the multiple testing that were carried out it will be important to clarify whether the p values were adjusted for multiple comparisons.

- We agree that adjusting for multiple comparisons is extremely important, particularly when sample sizes are relatively small. In the revised manuscript, we have substantially reduced the number of tests conducted by using multivariate analyses as recommended by R3. For our primary analyses (associations between the microbiome and fear reactivity), we use a Bonferroni correction which takes account of both the number of predictors (4 microbiome measures – Weighted Unifrac PC 1 & 2, Alpha PC 1 & 2) and the number of models that were run (2 models for the mask task, 2 for the strange situation paradigm, and 2 for the IBQ-R, with one model examining associations with the 1-month microbiome and one model examining associations with the 1-year microbiome). For our secondary analyses (associations between the microbiome and specific brain volumes), we also use a Bonferroni correction which takes account of both the number of predictors (4 microbiome measures) and the number of models that were run (1 model for associations between the 1-month microbiome and 1 month brain volumes, 1 model for associations between the 1-month microbiome and the 1-year brain volumes, and 1 model for associations between the 1-year microbiome and the 1-year brain volumes). Exploratory analyses testing for associations between individual genera and non-social fear reactivity are adjusted using FDR, as are our exploratory analyses testing for associations between individual genera and brain volumes.

Reviewer #2 (Remarks to the Author):

This is a fascinating and important investigation of links between the gut microbiota and fear behavior and amygdala volume in infants. There are many strengths of the study, including its longitudinal design, the methods chosen, and the analytic procedures. There are also some significant weaknesses of the study that require further consideration and elaboration.

First, the sample size is a major limitation, and although it is acknowledged by the authors, it gives me pause when examining behavioral measures that include as few as 14 infants. Although the behavioral measures used are among the best that are available for inferring fear behavior in infants, they are nevertheless complicated by large measurement error and basic difficulties, such as getting infants to sit in a highchair and attending to the task without crying (as experienced by the authors in this investigation). These difficulties are commonly experienced in this kind of work. I appreciate the careful exclusion of such infants in the current report along with the attention to interrater reliability. All these considerations are essential, but they do not compensate for the small sample size. Measuring behavior in infants is challenge and messy work and this is a major reason why a larger sample size is required to have confidence in the findings. Small sample sizes do not only suffer from type 1 errors, they are also prone to type 2 errors that arise for many reasons, including chance associations based on small error that can sometimes occur in small sample size. I also appreciate that the authors acknowledge the need for replication, but this is a requirement for all science, and the issue at hand is whether one can have confidence that the associations described here are in fact real. I also appreciate the framing of this study in terms of pilot work, nevertheless the focus is on hypothesis testing and p-values which means that it must be evaluated on this basis and not on the basis of effect sizes or other considerations that might be important to pilot work. My confidence in the conclusions would be increased with a larger sample size.

- We thank the reviewer for raising these important issues. The primary concern with small sample size is Type II error, a limitation which we acknowledge in the manuscript. In contrast, Type I error rate is dependent on the cutoff criterion for significance with statistical testing and not dependent on sample size. Our confidence in the predictor-outcome relationships is determined by the method of significance testing. The relationships we detail in the manuscript are highly significant both in previous models and now remain in multivariate mixed effect models. Multiple testing can inflate Type I error. To account for this, we used a conservative approach with stringent Bonferroni correction for main outcomes and FDR correction for exploratory analyses as detailed in the updated manuscript. Given the conservative cutoff set for significance in this work, we have confidence in these relationships despite the small sample size.

Second, some of my initial concerns about sample size were assuaged by the consistent negative association between the WU PC1 and amygdala volume at both 1-month and 1-year. On the surface at least, this consistency suggests reliable underlying processes. However, analysis of the change in microbiome from 1-month to 1-year reveals that the WU PC1 component is actually very different at 1-month versus 1-year. Specifically, at 1-month, WU PC1 has a positive association with Bifidobacterium and negative association with Bacteroides but at 1-year the pattern is exactly opposite. These 2 genera are obviously only 2 among many, so I don't want to extend the point too far, but from figure 4 we can see that at 1-month Bacteroides has the most

negative loading of any genera on WU PC1 whereas at 1-year it has the strongest positive loading on the same component. From a developmental perspective the change in composition of the microbiome may make sense, as described by the authors, but what requires further consideration is how the WU PC1 at these different timepoints with very different loading of genera like *Bacteroides* can have a consistently negative association with amygdala volume. I suspect that the answer to this puzzle lies in the developmental change that is described.

- WU PC1 at 1 month and WU PC1 at 1 year are not directly comparable as the principal coordinates are generated separately for each age group. We have clarified this in paragraph 1 of the revised results section. Each measure captures variation in the composition of the microbiome present at a particular age, and bacterial composition at 1 year is dramatically different than bacterial composition at 1 month (as the reviewer notes). The two PCs are not significantly correlated with each other (See Supplemental Table 2). Given the rapid pace of brain development in early life, and strong age-related changes in microbiome composition, it is quite likely that relationships between the microbiome and various phenotypes will be age-dependent, as they are in our data.

Third, the developmental change in the microbiome is not well integrated into the paper. It is described neither in the hypotheses nor the discussion. Furthermore, the justification for the 1-month and 1-year timepoints is lacking. Certainly, this kind of analysis may help to elucidate potentially confusing patterns of association as I describe above, but beyond that, such analysis could provide useful insight into developmental processes that may contribute to fear behavior in children. I find the developmental change described in figure 5 fascinating and potentially important to the focus of this paper, but this potential has not been realized.

- We selected the 1-month and 1-year timepoints because they index key periods in the development of the microbiome, brain, and fear behaviors. The 1-month sample captures an early point in the “developmental phase” of the infant gut microbiome as defined in large, population studies such as TEDDY (Stewart et al., Nature 2018). The 1-year sample captures the end of this same phase. Although the microbiome continues to change after this point, those changes are not as dramatic. Brain volumes show the greatest change from birth to 1 year of age. Finally, peak reactivity to fear-inducing stimuli occurs around the 1-year timepoint. We have added this information to page 6, paragraph 1 of the manuscript. We agree that future investigations would benefit from additional examination of longitudinal data. However, given reviewer concerns regarding multiple testing, we decided not to include additional analyses relating developmental changes in alpha diversity to behavioral and brain volume outcomes in this manuscript.

Fourth, I have a concern that some of the interpretation is not well justified by the data. For example, Fig 4 shows the taxa associated with each PC, but no data show whether these taxa are correlated with fear behavior or amygdala volume. The associations described are indirect: fear and amygdala to the PCs and PCs to taxa. However, the authors still made a connection between these taxa with fear behavior based on it. At minimum, this disconnect needs to be acknowledged.

- We thank the reviewer for the suggestion to include more information on genera-level associations with fear behavior and amygdala volume. We did conduct exploratory analyses testing the association of individual genera at 1 month and 1 year of age with brain and non-social fear outcomes. Results are described on pages 17 and 19 of the revised manuscript. Additional details are available in Table 2 and Supplemental Table 1. Although no relationships were significant after FDR correction, non-social fear behavior was associated ($p < 0.05$) with the relative abundance of *Dialister*, an unnamed genus in the Clostridiales order, members of the Clostridiales order that could not be confidently assigned to a family or genus, unnamed genus of Erysipelotrichaceae, and *Sutterella*, measured at 1 year of age. Three of these taxa (*Dialister*, members of the Clostridiales order that could not be confidently assigned to a family or genus, and the unnamed genus of Erysipelotrichaceae) also showed strong associations with 1-year Weighted Unifrac PC1. Regarding amygdala volume, there was a significant negative association between *Streptococcus* relative abundance at 1 month with amygdala volume at 1 month of age after FDR correction ($q = 0.021$) and *Streptococcus* is strongly associated with 1 month Weighted Unifrac PC1.

Additional minor points.

Line 59. The stimuli are not what emerges, but rather the response to them.

- This has been updated in the introduction to: “Fear behavior in response to different environmental stimuli emerge on a schedule that appears to parallel developmentally-relevant fitness threats, supporting an evolutionary non-associative model of fear acquisition.”

Line 61. Infants begin processing fear prior to 6-months of age, however their behavioral responses become reliable and strong enough by 6-months to allow their reliable measurement.

- We thank R2 for this clarification; the introduction text has been updated accordingly.

Line 73. Please indicate the direction of the effect.

- Direction of effect has been updated.

Line 76-77. As a general principle, the idea that timing of exposure to the microbiome is important to the acquisition of fear behavior does not make sense. The statement is certainly true of germ free mice. But all humans are exposed to microbiota from birth, and perhaps prior, thus the timing of exposure in humans means something rather different from GF mice.

- We agree that germ-free animal models are a unique experimental system that does not completely reflect biological reality. Never-the-less, these studies do suggest there are critical neurodevelopmental windows in which individual differences in composition of the microbiome may impact fear behavior acquisition and expression. Exposure to a certain type of microbiota during a period of rapid amygdala growth during the first year of life could be likened to that of any microbiota exposure in germ free animals during

this critical period. We have added this material to the introduction on page 4, paragraph 1.

Line 91, please indicate the direction of the effect.

- Direction of effect has been updated.

Line 121: The author mentioned that to reduce the confounding effect, one inclusion criteria is that all infants were breastfed until 1-month old. However, the influence of feeding practice to the gut microbiota can last until weaning age or even 1-year old. How is the feeding practice effect being adjusted for participants at 1-year age?

- We prospectively collected extensive feeding history from study participants through feeding recall during a phone interview at 6 months of age and at the study visit at 1 year of age (see Table 4 & Table 5). Feeding variables that demonstrated substantial variability between individuals were then tested for associations with community metrics of the infant gut microbiome. None of the feeding variables we examined were associated with these community metrics. This has been clarified in the revised manuscript (See “Identification of Microbiome Covariates”, page 22). This may seem surprising. However, recent research suggests that these factors may not have a large impact on community structure. In the TEDDY cohort (Stewart 2018), maternal BMI, birth mode, breast milk, solid food, probiotic, vitamin D, geographical location, household siblings, furry pets were all associated with genus level profiles, but only accounted for approximately 1-10% of the variance.

Line 161. It would be useful to add ‘1 year of age’ to the title for Table 1. I found myself a bit confused by what was measured at what timepoints, and this might help.

- Table has been updated to include age

Line 194. Given that the effect in figure 2 is not reliable, because it is known that it is driven by a single individual, it should be eliminated from the paper. Ensuring that the data is appropriate for analysis is a step prior to analysis, and findings with uncleaned data should not be reported. I do understand in this context that such findings may be intriguing because of the exploratory nature and because of the importance of not missing associations that may be present when the sample size is small, but in this case the effect is known to not be reliable because of the outlier and therefore should be reported as such.

- The updated statistical models with IBQ-R outcomes are no longer statistically significant. Consequently, this figure has been removed from the paper.

Line 286. Thank you for including Table 3, which provides important ‘background’ information. It does not relate to the hypotheses and should be moved to the supplemental material.

- This table has been moved to supplemental material as “Supplemental Table 2”

Line 291. The inclusion of covariates is important, and the included sensitivity analysis is important but not discussed. One is left wondering if vitamin D and maternal age really have anything to do with the associations or whether they are there by chance. It would be very useful to discuss the biological plausibility of these covariates in relation to the associations under investigation.

- We have clarified our approach to covariate identification in the results and methods. Updated statistical models now only show a positive significant relationship between maternal age at birth and 1-month Weighted Unifrac PC 2 (See page 22, paragraph 2). There were no significant associations of 1-month Weighted Unifrac PC 2 with outcomes of behavior or brain structure making sensitivity analyses unnecessary. While we acknowledge that the association of maternal age with the infant gut microbiome is interesting given that we know the gut microbiome changes as we age, we did not include additional discussion of this relationship in the revised manuscript as other larger prospective cohort studies examining development of the infant gut microbiome are better powered to address this relationship.

Lines 352 – 356. I am not clear on how studies on asthma and impairment of the immune system inform the suggestion on line 358ff. They don't seem to follow.

- With the updated statistical approach, it became clear that reduced evenness at 1 month of age was associated with increased fear behavior at 1 year of age rather than richness or alpha diversity in general. Evenness indexes the relative abundance of taxa in the microbiome and potentially speaks more to the production of metabolites by certain taxa. As such, our discussion has been updated to remove discussion about the immune system arm of the microbiome-gut-brain axis and includes new discussion of the need for future metabolomics investigations. See page 24, paragraph 2.

Line 425. I do agree on the strengths of the study, but the selection of subjects also should be addressed as a limitation to the generalizability and interpretation of the data.

- We thank R2 for this point and have updated the discussion accordingly on page 28, paragraph 1.

Line 631. Please include a description of the statistical model that was used. My interpretation is that a linear mixed model was used in which the repeated measures of fear behavior was used as an outcome, for each fear measure separately. This makes sense to me for the behavioral tasks because it can be broken down into multiple epochs but does not make sense to me regarding the IBQ R which only has one value. Please clarify. Also, please indicate which parameters were estimated in these models.

- We thank the reviewer for bringing this issue to our attention. We have updated the statistical analysis methods section to clarify our analysis approach. We have also rewritten and reorganized the results section to guide readers through the analysis. Linear mixed effect models with random intercept were used to test for effects of alpha and beta diversity on non-social fear behavior and social fear behavior. Multiple linear regression

models were used for IBQ-R fear index and brain volume outcomes. Alpha and beta diversity metrics, Alpha PC 1 & 2 and WU PC 1 and 2, were all included as multiple predictors in these models. Strange situation models of episodes 3.1, 3.2, and 3.3 used one-level mixed effect model to account for the within-subject covariance of successive episodes. For the mask task with both multiple correlated outcomes (facial fear, vocal distress, bodily fear, startle, escape behavior) and multiple episodes (up to 4 different masks presented), we adapted two-level random effects structure to account for the within-subject correlations among different mask task outcomes and within-subject but between-episode correlations of the same outcome. The parameters estimated are the fixed effect of diversity as predictors to the behavior outcomes.

Line 768. One of your key citations is incomplete, which prevents evaluating an important claim in the paper.

- Citation has been updated

Reviewer #3 (Remarks to the Author):

I wish to thank both the authors and the editors for the opportunity to review this manuscript. The authors are to be commended for their innovation in asking some very interesting questions, and the rigor of their behavioral data collection and coding.

In essence, the investigators use two measures (alpha diversity and taxonomic community structure) to link the infant gut microbiome development with behavior in the first year of life. They observe significant coarse correlations, which are both interesting and novel for human cohort studies. However, there are also several major concerns.

1. There is an inverse relationship between alpha diversity and social fear, which held true at several relative abundance measures of beta diversity but did not hold true in inferred metagenomics analyses.

I find this troubling for two reasons. First, decreased alpha diversity is most related to breast milk and specifically mothers own milk feeding. The investigators seem to suggest that this diminished alpha diversity is a "bad thing" ergo leading to social fears. However, this could simply be a pathway to adjustment.

- At 1 month of age, all of the subjects in this study were exclusively breastfed. Consequently, our results cannot be explained by differences in feeding practice (breast versus formula). However, it is possible that the composition of breastmilk might influence 1-month microbiome diversity. We have added a brief discussion of this possibility to the main text (see page 26, paragraph 2). Given that research on the infant microbiome and behavioral and health outcomes is still in the discovery phase, we felt it would be premature to label certain microbiome metrics as “good” or “bad”. Indeed, it may be the case that microbiomes that are good for one outcome are bad for other outcomes. In regard to fear behavior, we know that extremes of fear behavior at 1 year of age are linked to future anxiety-disorders (high reactivity) and callous-unemotional traits (low reactivity) and this is noted in the manuscript. However, fear behavior itself is a normal part of development. Rather than assign pathologic significance in this manuscript, we focused on the novel association of the human infant microbiome with variation in fear behavior as previously described in preclinical animal models. Discussion of this has been added as a limitation on page 28, paragraph 1.

It would be helpful to more formally integrate with GLM models (or proc model) the mode and manner of feeding with the fear, and then do a multipartite analysis with both alpha and beta diversity.

- We thank the reviewer for the excellent suggestion to incorporate both alpha and beta diversity as predictors in our models, and have updated our analysis accordingly. Please see page 40 for full description of updated statistical models. We did not incorporate mode and manner of feeding into the primary analyses, as

none of the feeding variables we examined were associated with the community metrics (See “Identification of Microbiome Covariates” on page 22).

Second, I find it concerning that there is no functional metagenomics relationship. While this may be due to the coarseness and inference approach they used (PICRUST), it is crucially important to discuss and attempt to further understand. In other words, if the pathways don't actually differ, are they only measuring small but non-relevant community membership differences?

- We thank the reviewer for bringing up this important point. The lack of relationships observed between beta diversity and predicted metagenomic functional ability may be explained by the limitations of PICRUST or the functional redundancy of genomic potential across different microbiome taxa. First, PICRUST is limited because it does not contain genomic information for all strains identified in our sample. This missing data could account for the lack of relationships between predicted function and our beta diversity metrics. Furthermore, PICRUST cannot capture differences in microbial gene expression; it can only capture predicted functional capacity. While there may not be significant associations with the predicted metagenomic functional ability in this data, there could be differential expression of microbial genes in samples on opposite ends of Weighted Unifrac PC1. Future transcriptomic studies could clarify this issue. Discussion of this has been added to page 24, paragraph 1.

2. The initial correlation matrices don't seem to ever be fully resolved in the multidimensional analyses. Perhaps I am missing some key aspects to their analysis, but it would appear that (as anticipated) there is significant interaction among behaviors. This is certainly expected. However, it is less clear how and if they looked for these same interactions as explanatory of these rather small microbiome alpha diversity measures.

- We thank the reviewer for bringing this issue to our attention. We have updated our statistical analysis to better reflect the relationships revealed by the correlation matrices. Specifically, the correlation matrices indicate that measures of facial fear, bodily fear, vocal distress, escape behavior, and startle within the Mask Task paradigm were highly correlated, suggesting they tap into the same underlying behavioral construct. However, there was little correlation between Mask Task, Strange Situation, and IBQ-R behavioral assessments, suggesting these measures capture different behavioral constructs. Consequently, we ran separate models for the Mask Task, Strange Situation, and IBQ-R. Because the Mask Task has both multiple correlated outcomes (facial fear, vocal distress, bodily fear, startle, escape behavior) and multiple episodes (up to 4 different masks presented), we adapted a two-level random effects structure to account for the within-subject correlations among different mask task outcomes and within-subject but between-episode correlations of the same outcome, when testing for associations with alpha and beta diversity measures. We have updated the statistical analysis methods section to clarify our analysis approach. We have also rewritten and reorganized the results section to guide readers through the analysis.

3. A real lost opportunity with the study in its current form is the absence of MDD or Dirilecht modeling over time (i.e., longitudinal measures of community composition and change). In other words, is there a difference in the stability or dynamic nature of the maturation of the communities that correlates/corresponds with fear responses? There are several good ways to model and measure with the gut microbiome, and this would be important to look at.

- We agree that DMM modeling can reveal interesting information about changes in community composition over time. However, DMM longitudinal modeling would not be appropriate for our dataset as samples at 1 month and 1 year of age were sequenced separately. Furthermore, DMM works best in large cohorts, where many samples are collected from each subject over a substantial period of time (as in the TEDDY study). Although we could have applied clustering approaches separately to the 1-month and 1-year data, and then looked at changes in cluster membership between the two time points, the resulting cell size would be very small and subsequent analyses very underpowered. As noted in our response to R2, we could have examined changes in alpha diversity over time in relation to behavior and brain outcomes. However, given reviewer concerns regarding multiple testing, we decided not to include additional analyses relating developmental changes in alpha diversity to behavioral and brain volume outcomes in this manuscript.

Reviewer #4 (Remarks to the Author):

Summary

This study investigates the gut microbiome composition and diversity at 1 month and 1 year of age, and associates these with fear-related behavior and volumes of amygdala, hippocampus and medial prefrontal cortex, also measured at 1 month and 1 year of age. In the 1-month associations there is around $n=19$ for the behavioral associations and $n=27$ for brain volume associations, and at 1 year, there are around $n=14$ for the behavioral associations and $n=13$ for the brain volume associations. There were negative associations between microbiome diversity and fear at 1 month, and with fear and microbiome composition at 1 year. There was a negative association between 1 month (and 1 year) microbiome diversity and 1 month amygdala volume, but positive association with 1 year amygdala volume. The authors conclude that they are first to demonstrate associations of gut microbiome with fear behavior and the amygdala (a fear-related brain structure).

This is a generally well-written manuscript with a novel and interesting topic. My main concern is that the numbers used in this study are very small, and therefore we cannot be sure that the effects reported are real. The associations and their direction seem a bit random, and may be due to multiple comparisons, outliers, or confounders that I'm not sure are properly dealt with.

- We thank the reviewer for raising these important issues. The primary concern with small sample size is Type II error, a limitation which we acknowledge in the manuscript. In contrast, Type I error rate is dependent on the cutoff criterion for significance with statistical testing and not dependent on sample size. Our confidence in the predictor-outcome relationships is determined by the method of significance testing. The relationships we detail in the manuscript are highly significant both in previous models and now remain in multivariate mixed effect models. Multiple testing can inflate Type I error. To account for this, we used a conservative approach with stringent Bonferroni correction for main outcomes and FDR correction for exploratory analyses as detailed in the updated manuscript. Given the conservative cutoff set for significance in this work, we have confidence in these relationships despite the small sample size. We did consider a wide-range of potential confounders and have updated the manuscript to clarify this (See "Identification of Microbiome Covariates" on page 22). Regarding outliers, we display individual data points for our primary and secondary analyses (see Figures 3, 4, and 5) and did not observe any influential data points.

Abstract

An idea of the numbers of data for each analysis at each timepoint would be helpful to the reader.

- Abstract has been updated to reflect this suggestion

Introduction

The introduction is excellent and motivates the study very well.

Results

It is unclear whether the results reported in figure 1-3 are corrected for multiple comparisons?

- We agree that adjusting for multiple comparisons is extremely important, particularly when sample sizes are relatively small. In the revised manuscript, we have substantially reduced the number of tests conducted by using multivariate analyses as recommended by R3. For our primary analyses (associations between the microbiome and fear reactivity), we use a Bonferroni correction which takes account of both the number of predictors (4 microbiome measures – Weighted Unifrac PC 1 & 2, Alpha PC 1 & 2) and the number of models that were run (2 models for the mask task, 2 for the strange situation paradigm, and 2 for the IBQ-R, with one model examining associations with the 1-month microbiome and one model examining associations with the 1-year microbiome). For our secondary analyses (associations between the microbiome and specific brain volumes), we also use a Bonferroni correction which takes account of both the number of predictors (4 microbiome measures) and the number of models that were run (1 model for associations between the 1-month microbiome and 1 month brain volumes, 1 model for associations between the 1-month microbiome and the 1-year brain volumes, and 1 model for associations between the 1-year microbiome and the 1-year brain volumes). Exploratory analyses testing for associations between individual genera and non-social fear reactivity are adjusted using FDR, as are our exploratory analyses testing for associations between individual genera and brain volumes.

There are many instances where the direction of the results is not clearly stated.

- Direction of effect has been updated throughout the text.

The number of abbreviations makes it difficult for the reader to follow.

- Abbreviations have been minimized throughout the manuscript for clarity

Has KEGG abbreviation been defined?

- KEGG was defined in the results

Again, the direction of the associations is sometimes not stated in the ‘identification of covariates’ section.

- Direction of effect has been updated throughout the text.

I would have thought you would add both identified covariates to the sensitivity analyses together?

- Updated statistical models now only show a positive significant relationship between maternal age at birth and 1-month Weighted Unifrac PC 2 (See page 22, paragraph 2). There were no significant associations of 1-month Weighted Unifrac PC 2 with outcomes of behavior or brain structure making sensitivity analyses unnecessary.

I am concerned that some of the covariates that were not found to be ‘significantly associated’ with the microbiome or behavioral measures may still be important (especially since multiple comparisons correction was applied here, but not for the main analysis, if I understand correctly?) Potential influencers of brain volume are not considered for those analyses, it is well known that sex and age at scan affects brain volume, for instance.

- We prospectively collected extensive feeding history from study participants through feeding recall during a phone interview at 6 months of age and at the study visit at 1 year of age (see Table 4 & Table 5). Feeding variables that demonstrated substantial variability between individuals were then tested for associations with community metrics of the infant gut microbiome. None of the feeding variables we examined were associated with these community metrics. This has been clarified in the revised manuscript (See “Identification of Microbiome Covariates”, page 22). This may seem surprising. However, recent research suggests that these factors may not have a large impact on community structure. In the TEDDY cohort (Stewart 2018), maternal BMI, birth mode, breast milk, solid food, probiotic, vitamin D, geographical location, household siblings, furry pets were all associated with genus level profiles, but only accounted for approximately 1-10% of the variance. We agree that age at scan and sex are important covariates for brain volumes. All analyses with brain volumes have been rerun to include these covariates. The brain volume results remain largely the same. See page 17, paragraph 3.

It is fine that you chose particular brain regions hypothesized to be involved in fear behavior, but I also wonder if you would find random other associations with other brain regions, i.e. would these brain regions be uniquely and specifically associated with the microbiome?

- We limited our analysis to areas involved in fear behavior as stated in our *a priori* hypotheses based on a number of studies conducted in animal models. Manipulation of the microbiome has been shown to impact the amygdala, hippocampus, and mPFC areas. The small sample size in this study does not lend itself to exploratory analysis of all gray matter areas in our parcellation. This work supports future studies powered for this type of analysis.

Discussion

It was difficult to get an idea of the main interpretation of the findings from the discussion. I found it hard to understand which direction of the microbiome metrics were ‘good’ or ‘bad’ for instance, and likewise for the species composition.

- Given that research on the infant microbiome and behavioral and health outcomes is still in the discovery phase, we felt it would be premature to label certain microbiome metrics as “good” or “bad”. Indeed, it may be the case that microbiomes that are good for one outcome are bad for other outcomes. In regard to fear behavior, we know that extremes of fear behavior at 1 year of age are linked to future anxiety-disorders (high reactivity) and callous-unemotional traits (low reactivity) and this is noted in the manuscript. However, fear behavior itself is a normal part of development. Rather than assign pathologic significance in this manuscript, we focused on the novel association of the human infant microbiome with variation in fear behavior as previously described in preclinical animal models. Discussion of this has been added as a limitation on page 28, paragraph 1.

The discussion of results seems a bit selective, based on how the narrative is framed by the authors. There is not much discussion of the results that remained from the sensitivity analyses, for instance.

- Regarding sensitivity analysis discussion, updated statistical models now only show a positive significant relationship between maternal age at birth and 1-month Weighted Unifrac PC 2 (See page 22, paragraph 2). There were no significant associations of 1-month Weighted Unifrac PC 2 with outcomes of behavior or brain structure making sensitivity analyses unnecessary. While we acknowledge that the association of maternal age with the infant gut microbiome is interesting given that we know the gut microbiome changes as we age, we did not include additional discussion of this relationship in the revised manuscript as other larger prospective cohort studies examining development of the infant gut microbiome are better powered to address this relationship.

Also, line 403-410 doesn't discuss the 1-month WU associations (e.g. with 1-year amygdala, which was the only one significant after covariate adjustment).

- With new models, community metrics associated with brain volumes do not survive Bonferroni correction. The discussion has been updated to reflect this. We also added discussion of exploratory analyses related individual genera to brain volumes, where *Streptococcus* relative abundance was significantly associated with 1-month amygdala volume after FDR correction (see page 26).

I'm not sure of the relevance of some of the inflammation discussion regarding linking microbes and anxiety.

- With the updated statistical approach, it became clear that reduced evenness at 1 month of age was associated with increased fear behavior at 1 year of age rather than richness or alpha diversity in general. Evenness indexes the relative abundance of taxa in the microbiome and potentially speaks more to the production of metabolites by certain taxa. As such, our discussion has been updated to remove discussion about the immune system arm of the microbiome-gut-brain axis and includes new discussion of the need for future metabolomics investigations. See page 24, paragraph 2.

The conclusion is good in that it does not overstate these findings.

Methods

It is good that the inclusion criteria removed some of the potential confounders at 1 month, but one wonders about their confounding effects later. It is difficult to deal with these properly, since the sample size is so small.

- We collected detailed information about sociodemographic, medical, and feeding variables at 1 month, 6 months, and 1 year of age. Few associations with potential confounders were seen within our data. Additional discussion of sensitivity analyses has been added to the manuscript. While the microbiome has been shown to be impacted by a variety of medical/feeding/demographic factors, the effect sizes in large cohort studies are surprisingly small. In the TEDDY cohort at 1 year of age, the microbiome was associated with only breast milk, probiotic, geographic location, and household siblings out of many variables tested with less than 5% of the variance was explained by these factors (Stewart 2018).

Please state the range of age at each visit.

- The range of age at each visit has been updated in methods.

Typo 'at' on line 548.

- Addressed

The sequencing details seem comprehensive, but I am not qualified to comment on this methodology!

Is the Gousias template for 2-year-olds? It might be worth discussing potential issues with using this on 1 month and 1-year MRI data.

- The template is based on the Gousias multi-atlas template of 33 typically developing kids at age 2 years (Gousias 2008). Using a multi-atlas fusion procedure (Wang 2014) we propagated the Gousias parcellation to the MNI 1-year pediatric template space. The propagated parcellation was visually assessed and found to be of high anatomical accuracy in the 1 year pediatric T1w and T2w template images. In order to ensure a consistent parcellation in both neonate and 1-year-old setting, we applied this 1-year pediatric template to all datasets independent of age. Finally, all parcellation results were visually assessed for appropriate parcellation performance. The methods have been updated to reflect this clarification. See page 39, paragraph 2.

The statistical methods section is a bit sparse, and in particular does not mention correction for multiple comparisons.

- We thank the reviewer for bringing this issue to our attention. We have updated the statistical analysis methods section to clarify our analysis approach. We have also rewritten and reorganized the results section to guide readers through the analysis. Linear

mixed effect models with random intercept were used to test for effects of alpha and beta diversity on non-social fear behavior and social fear behavior. Multiple linear regression models were used for IBQ-R fear index and brain volume outcomes. Alpha and beta diversity metrics, Alpha PC 1 & 2 and WU PC 1 and 2, were all included as multiple predictors in these models. Strange situation models of episodes 3.1, 3.2, and 3.3 used one-level mixed effect model to account for the within-subject covariance of successive episodes. For the mask task with both multiple correlated outcomes (facial fear, vocal distress, bodily fear, startle, escape behavior) and multiple episodes (up to 4 different masks presented), we adapted two-level random effects structure to account for the within-subject correlations among different mask task outcomes and within-subject but between-episode correlations of the same outcome. The parameters estimated are the fixed effect of diversity as predictors to the behavior outcomes. We agree that adjusting for multiple comparisons is extremely important, particularly when sample sizes are relatively small. In the revised manuscript, we have substantially reduced the number of tests conducted by using multivariate analyses as recommended by R3. For our primary analyses (associations between the microbiome and fear reactivity), we use a Bonferroni correction which takes account of both the number of predictors (4 microbiome measures – Weighted Unifrac PC 1 & 2, Alpha PC 1 & 2) and the number of models that were run (2 models for the mask task, 2 for the strange situation paradigm, and 2 for the IBQ-R, with one model examining associations with the 1-month microbiome and one model examining associations with the 1-year microbiome). For our secondary analyses (associations between the microbiome and specific brain volumes), we also use a Bonferroni correction which takes account of both the number of predictors (4 microbiome measures) and the number of models that were run (1 model for associations between the 1-month microbiome and 1 month brain volumes, 1 model for associations between the 1-month microbiome and the 1-year brain volumes, and 1 model for associations between the 1-year microbiome and the 1-year brain volumes). Exploratory analyses testing for associations between individual genera and non-social fear reactivity are adjusted using FDR, as are our exploratory analyses testing for associations between individual genera and brain volumes.

Reviewer #2 (Remarks to the Author):

The authors have been responsive to the concerns raised in my initial review. The clarifications have been helpful in more fully evaluating the work. I remain concerned about the small sample size and the fact that statistical inferences are being made on a very small sample size. The authors note that type 1 errors are not affected by sample size, however any factor that affects standard errors also will affect p values, and small samples can sometimes be surprisingly homogeneous leading to statistical inferences that would not be made on a sample that properly represents the population to which the inferences are made. Very small sample sizes also tend to lead to unstable results. It is reassuring when data analyzed in different ways obtained the same results. I realize that different statistical approaches also make different assumptions and this can be a reason for differences in results, however I worry that differences in findings between the original submission and the current version may reflect a lack of stability in the results. To somewhat offset this concern, the authors had a priori hypotheses and have corrected for multiple comparison. I wonder why the authors have not collected additional data to increase the sample size? I think I would be more comfortable with the manuscript if the title and the abstract contained the word 'pilot' as a way to alert the reader to the fact that these are preliminary results.

Reviewer #3 (Remarks to the Author):

I wish to commend the authors for their highly responsive revised submission. I feel the concerns by the reviewers have been fully addressed.

Reviewer #4 (Remarks to the Author):

The authors have clearly put a lot of effort in to addressing the reviewer comments, and I have no further suggestions.

Response to Review:

Reviewer #2 (Remarks to the Author):

1. The authors have been responsive to the concerns raised in my initial review. The clarifications have been helpful in more fully evaluating the work. I remain concerned about the small sample size and the fact that statistical inferences are being made on a very small sample size. The authors note that type 1 errors are not affected by sample size, however any factor that affects standard errors also will affect p values, and small samples can sometimes be surprisingly homogeneous leading to statistical inferences that would not be made on a sample that properly represents the population to which the inferences are made. Very small sample sizes also tend to lead to unstable results. It is reassuring when data analyzed in different ways obtained the same results. I realize that different statistical approaches also make different assumptions and this can be a reason for differences in results, however I worry that differences in findings between the original submission and the current version may reflect a lack of stability in the results. To somewhat offset this concern, the authors had a priori hypotheses and have corrected for multiple comparison. I wonder why the authors have not collected additional data to increase the sample size? I think I would be more comfortable with the manuscript if the title and the abstract contained the word 'pilot' as a way to alert the reader to the fact that these are preliminary results.
 - We thank the reviewer for these additional comments regarding the relatively small sample size of the study. We have made several changes in response. First, we changed the title as the reviewer suggested; it now reads “Infant Gut Microbiome Associated with Fear Behavior in a Pilot Study”. Second, we have revised the abstract to clearly state that our findings are based on a small cohort and that the study “requires further validation with a larger number of participants”. Finally, we have added some additional information to page 29 of the manuscript (Discussion section) regarding making statistical inferences in a small sample. Specifically, we state that “Small sample sizes can produce unstable results and we acknowledge there is the risk of homogenous sampling leading to statistical inferences that do not represent the overall population.”

Reviewer #3 (Remarks to the Author):

2. I wish to commend the authors for their highly responsive revised submission. I feel the concerns by the reviewers have been fully addressed.
 - We thank the reviewer for their positive comments.

Reviewer #4 (Remarks to the Author):

3. The authors have clearly put a lot of effort in to addressing the reviewer comments, and I have no further suggestions.
 - We thank the reviewer for their positive comments.